# Learning Multimodal VAEs through Mutual Supervision

**Tom Joy[1], Yuge Shi[1], Philip H.S. Torr[1], Tom Rainforth[1], Sebastian Schmon[2], N. Siddharth[3]**

[1]University of Oxford
[2]University of Durham
[3]University of Edinburgh & The Alan Turing Institute
`tomjoy@robots.ox.ac.uk`

## Abstract

Multimodal variational autoencoders (VAEs) seek to model the joint distribution over heterogeneous data (e.g. vision, language), whilst also capturing a shared representation across such modalities. Prior work has typically combined information from the modalities by reconciling idiosyncratic representations directly in the recognition model through explicit products, mixtures, or other such factorisations. Here we introduce a novel alternative, the **M**utually sup**E**rvised **M**ultimodal VA**E** (MEME), that avoids such explicit combinations by repurposing semi-supervised VAEs to combine information between modalities *implicitly* through mutual supervision. This formulation naturally allows learning from partially-observed data where some modalities can be entirely missing—something that most existing approaches either cannot handle, or do so to a limited extent. We demonstrate that MEME outperforms baselines on standard metrics across *both* partial and complete observation schemes on the MNIST-SVHN (image–image) and CUB (image–text) datasets[1]. We also contrast the quality of the representations learnt by mutual supervision against standard approaches and observe interesting trends in its ability to capture relatedness between data.

## 1 Introduction

Modelling the generative process underlying heterogenous data, particularly data spanning multiple perceptual modalities such as vision or language, can be enormously challenging. Consider for example, the case where data spans across photographs and sketches of objects. Here, a data point, comprising of an instance from each modality, is constrained by the fact that the instances are related and must depict the *same* underlying abstract concept. An effective model not only needs to faithfully generate data in each of the different modalities, it also needs to do so in a manner that preserves the underlying relation between modalities. Learning a model over multimodal data thus relies on the ability to bring together information from idiosyncratic sources in such a way as to overlap on aspects they relate on, while remaining disjoint otherwise.

Variational autoencoders (VAEs) (Kingma & Welling, 2014) are a class of deep generative models that are particularly well-suited for multimodal data as they employ the use of *encoders*—learnable mappings from high-dimensional data to lower-dimensional representations—that provide the means to combine information across modalities. They can also be adapted to work in situations where instances are missing for some modalities; a common problem where there are inherent difficulties in obtaining and curating heterogenous data. Much of the work in multimodal VAEs involves exploring different ways to model and formalise the combination of information with a view to improving the quality of the learnt models (see § 2).

Prior approaches typically combine information through *explicit* specification as products (Wu & Goodman, 2018), mixtures (Shi et al., 2019), combinations of such (Sutter et al., 2021), or through additional regularisers on the representations (Suzuki et al., 2016; Sutter et al., 2020). Here, we explore an alternative approach that leverages advances in semi-supervised VAEs (Siddharth et al.,

---

[1]The codebase is available at the following location: https://github.com/thwjoy/meme.

Figure 1: Constraints on the representations. *(a) VAE*: A prior regularises the data encoding distribution through KL. *(b) Typical multimodal VAE*: Encodings for different modalities are first explicitly combined, with the result regularised by a prior through KL. *(c) MEME (ours)*: Leverage semi-supervised VAEs to cast one modality as a conditional prior, implicitly supervising/regularising the other through the VAE's KL. Mirroring the arrangement to account for KL asymmetry enables multimodal VAEs through mutual supervision.

2017; Joy et al., 2021) to repurpose existing regularisation in the VAE framework as an *implicit* means by which information is combined across modalities (see Figure 1).

We develop a novel formulation for multimodal VAEs that views the combination of information through a semi-supervised lens, as *mutual supervision* between modalities. We term this approach **M**utually sup**E**rvised **M**ultimodal VA**E** (MEME). Our approach not only avoids the need for additional explicit combinations, but it also naturally extends to *learning* in the partially-observed setting—something that most prior approaches cannot handle. We evaluate MEME on standard metrics for multimodal VAEs across both partial *and* complete data settings, on the typical multimodal data domains, MNIST-SVHN (image-image) and the less common but notably more complex CUB (image-text), and show that it outperforms prior work on both. We additionally investigate the capability of MEMEs ability to capture the 'relatedness', a notion of semantic similarity, between modalities in the latent representation; in this setting we also find that MEME outperforms prior work considerably.

## 2 RELATED WORK

Prior approaches to multimodal VAEs can be broadly categorised in terms of the explicit combination of representations (distributions), namely concatenation and factorization.

**Concatenation**: Models in this category learn joint representation by either concatenating the inputs themselves or their modality-specific representations. Examples for the former includes early work in multimodal VAEs such as the JMVAE (Suzuki et al., 2016), triple ELBO (Vedantam et al., 2018) and MFM (Tsai et al., 2019), which define a joint encoder over concatenated multimodal data. Such approaches usually require the training of auxiliary modality-specific components to handle the partially-observed setting, with missing modalities, at test time. They also cannot learn from partially-observed data. In very recent work, Gong et al. (2021) propose VSAE where the latent representation is constructed as the concatenation of modality-specific encoders. Inspired by VAEs that deal with imputing pixels in images such as VAEAC (Ivanov et al., 2019), Partial VAE (Ma et al., 2018), MIWAE (Mattei & Frellsen, 2019), HI-VAE (Nazábal et al., 2020) and pattern-set mixture model (Ghalebikesabi et al., 2021), VSAE can learn in the partially-observed setting by incorporating a modality mask. This, however, introduces additional components such as a collective proposal network and a mask generative network, while ignoring the need for the joint distribution over data to capture some notion of the relatedness between modalities.

**Factorization**: In order to handle missing data at test time without auxiliary components, recent work propose to factorize the posterior over all modalities as the product (Wu & Goodman, 2018) or mixture (Shi et al., 2019) of modality-specific posteriors (experts). Following this, Sutter et al. (2021) proposes to combine the two approaches (MoPoE-VAE) to improve learning in settings where the number of modalities exceeds two. In contrast to these methods, mmJSD (Sutter et al., 2020) combines information not in the posterior, but in a "dynamic prior", defined as a function (either mixture or product) over the modality-specific posteriors as well as pre-defined prior.

Table 1 provides a high-level summary of prior work. Note that all the prior approaches have some explicit form of joint representation or distribution, where some of them induces the need for auxiliary components to deal with missing data at test time, while others are established without significant theoretical benefits. By building upon a semi-supervised framework, our method MEME circumvents this issue to learn representations through mutual supervision between modalities, and is able to deal with missing data at train or test time naturally without additional components.

Table 1: We examine four characteristics: The ability to handle partial observation at test and train time, the form of the joint distribution or representation in the bi-modal case (**s**, **t** are modalities), and additional components. (✓) indicates a theoretical capability that is not verified empirically.

| | Partial Test | Partial Train | Joint repr./dist. | Additional |
|---|---|---|---|---|
| JMVAE | ✓ | ✗ | $q_\Phi(\mathbf{z}|\mathbf{s},\mathbf{t})$ | $q_{\phi_s}(\mathbf{z}|\mathbf{s}), q_{\phi_t}(\mathbf{z}|\mathbf{t})$ |
| tELBO | ✓ | ✗ | $q_\Phi(\mathbf{z}|\mathbf{s},\mathbf{t})$ | $q_{\phi_s}(\mathbf{z}|\mathbf{s}), q_{\phi_t}(\mathbf{z}|\mathbf{t})$ |
| MFM | ✓ | ✗ | $q_\Phi(\mathbf{z}|\mathbf{s},\mathbf{t})$ | $q_{\phi_s}(\mathbf{z}|\mathbf{s}), q_{\phi_t}(\mathbf{z}|\mathbf{t})$ |
| VSVAE | ✓ | ✓ | $\texttt{concat}(z_s, z_t)$ | mask generative network |
| MVAE | ✓ | (✓) | $q_{\phi_s}(\mathbf{z}|\mathbf{s})q_{\phi_t}(\mathbf{z}|\mathbf{t})p(\mathbf{z})$ | sub-sampling |
| MMVAE | ✓ | ✗ | $q_{\phi_s}(\mathbf{z}|\mathbf{s}) + q_{\phi_t}(\mathbf{z}|\mathbf{t})$ | - |
| MoPoE | ✓ | (✓) | $q_{\phi_s}(\mathbf{z}|\mathbf{s}) + q_{\phi_t}(\mathbf{z}|\mathbf{t}) + q_{\phi_s}(\mathbf{z}|\mathbf{s})q_{\phi_t}(\mathbf{z}|\mathbf{t})$ | - |
| mmJSD | ✓ | ✗ | $f(q_{\phi_s}(\mathbf{z}|\mathbf{s}), q_{\phi_t}(\mathbf{z}|\mathbf{t}), p(\mathbf{z}))$ | - |
| **Ours** | ✓ | ✓ | - | - |

## 3 METHOD

Consider a scenario where we are given data spanning two modalities, **s** and **t**, curated as pairs (**s**, **t**). For example this could be an "image" and associated "caption" of an observed scene. We will further assume that some proportion of observations have one of the modalities missing, leaving us with partially-observed data. Using $\mathcal{D}_{\mathbf{s},\mathbf{t}}$ to denote the proportion containing fully observed pairs from both modalities, and $\mathcal{D}_\mathbf{s}$, $\mathcal{D}_\mathbf{t}$ for the proportion containing observations only from modality **s** and **t** respectively, we can decompose the data as $\mathcal{D} = \mathcal{D}_\mathbf{s} \cup \mathcal{D}_\mathbf{t} \cup \mathcal{D}_{\mathbf{s},\mathbf{t}}$.

In aid of clarity, we will introduce our method by confining attention to this bi-modal case, providing a discussion on generalising beyond two modalities later. Following established notation in the literature on VAEs, we will denote the generative model using $p$, latent variable using $\mathbf{z}$, and the encoder, or recognition model, using $q$. Subscripts for the generative and recognition models, where indicated, denote the parameters of deep neural networks associated with that model.

### 3.1 SEMI-SUPERVISED VAEs

To develop our approach we draw inspiration from semi-supervised VAEs which use additional information, typically data labels, to extend the generative model. This facilitates learning tasks such as disentangling latent representations and performing intervention through conditional generation. In particular, we will build upon the work of Joy et al. (2021), who suggests to supervise latent representations in VAEs with partial label information by forcing the encoder, or recognition model, to channel the flow of information as $\mathbf{s} \rightarrow \mathbf{z} \rightarrow \mathbf{t}$. They demonstrate that the model learns latent representations, $\mathbf{z}$, of data, $\mathbf{s}$, that can be faithfully identified with label information $\mathbf{t}$.

Figure 2 shows a modified version of the graphical model from Joy et al. (2021), extracting just the salient components, and avoiding additional constraints therein. The label, here **t**, is denoted as partially observed as not all observations **s** have associated labels. Note that, following the information flow argument, the generative model factorises as $p_{\theta,\psi}(\mathbf{s},\mathbf{z},\mathbf{t}) = p_\theta(\mathbf{s} \mid \mathbf{z})\, p_\psi(\mathbf{z} \mid \mathbf{t})\, p(\mathbf{t})$ (solid arrows) whereas the recognition model factorises as $q_{\phi,\varphi}(\mathbf{t},\mathbf{z} \mid \mathbf{s}) = q_\varphi(\mathbf{t} \mid \mathbf{z})\, q_\phi(\mathbf{z} \mid \mathbf{s})$ (dashed arrows). This autoregressive formulation of both the generative and recognition models is what enables the "supervision" of the latent representation of **s** by the label, **t**, via the conditional prior $p_\psi(\mathbf{z} \mid \mathbf{t})$ as well as the classifier $q_\varphi(\mathbf{t} \mid \mathbf{z})$.

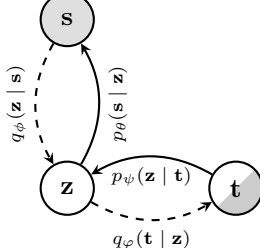

Figure 2: Simplified graphical model from Joy et al. (2021).

The corresponding objective for *supervised* data, derived as the (negative) variational free energy or evidence lower bound (ELBO) of the model is

$$\log p_{\theta,\psi}(\mathbf{s},\mathbf{t}) \geq \mathcal{L}_{\{\boldsymbol{\Theta},\boldsymbol{\Phi}\}}(\mathbf{s},\mathbf{t}) = \mathbb{E}_{q_\phi(\mathbf{z}|\mathbf{s})}\left[\frac{q_\varphi(\mathbf{t}|\mathbf{z})}{q_\phi(\mathbf{z}|\mathbf{s})}\log\frac{p_\theta(\mathbf{s}|\mathbf{z})p_\psi(\mathbf{z}|\mathbf{t})}{q_\phi(\mathbf{z}|\mathbf{s})q_\varphi(\mathbf{t}|\mathbf{z})}\right] + \log q_{\phi,\varphi}(\mathbf{t}|\mathbf{s}) + \log p(\mathbf{t}), \quad (1)$$

with the generative and recognition model parameterised by $\boldsymbol{\Theta} = \{\theta, \vartheta\}$ and $\boldsymbol{\Phi} = \{\phi, \varphi\}$ respectively. A derivation of this objective can be found in Appendix A.

## 3.2 Mutual Supervision

Procedurally, a semi-supervised VAE is already multimodal. Beyond viewing labels as a separate data modality, for more typical multimodal data (vision, language), one would just need to replace labels with data from the appropriate modality, and adjust the corresponding encoder and decoder to handle such data. Conceptually however, this simple replacement can be problematic.

Supervised learning encapsulates a very specific imbalance in information between observed data and the labels—that labels do not encode information beyond what is available in the observation itself. This is a consequence of the fact that labels are typically characterised as projections of the data into some lower-dimensional conceptual subspace such as the set of object classes one may encounter in images, for example. Such projections cannot introduce additional information into the system, implying that the information in the data subsumes the information in the labels, i.e. that the conditional entropy of label $\mathbf{t}$ given data $\mathbf{s}$ is zero: $H(\mathbf{t} \mid \mathbf{s}) = 0$. Supervision-based models typically incorporate this information imbalance as a feature, as observed in the specific correspondences and structuring enforced between their label $\mathbf{y}$ and latent $\mathbf{z}$ in Joy et al. (2021).

Multimodal data of the kind considered here, on the other hand, does not exhibit this feature. Rather than being characterised as a projection from one modality to another, they are better understood as idiosyncratic projections of an abstract concept into distinct modalities—for example, as an image of a bird or a textual description of it. In this setting, no one modality has *all* the information, as each modality can encode unique perspectives opaque to the other. More formally, this implies that both the conditional entropies $H(\mathbf{t} \mid \mathbf{s})$ and $H(\mathbf{s} \mid \mathbf{t})$ are finite.

Based on this insight we symmetrise the semi-supervised VAE formulation by additionally constructing a mirrored version, where we swap $\mathbf{s}$ and $\mathbf{t}$ along with their corresponding parameters, i.e. the generative model now uses the parameters $\mathbf{\Phi}$ and the recognition model now uses the parameters $\mathbf{\Theta}$. This has the effect of also incorporating the information flow in the opposite direction to the standard case as $\mathbf{t} \rightarrow \mathbf{z} \rightarrow \mathbf{s}$, ensuring that the modalities are now *mutually supervised*. This approach forces each encoder to act as an encoding distribution when information flows one way, but also act as a prior distribution when the information flows the other way. Extending the semi-supervised VAE objective (6), we construct a bi-directional objective for MEME

$$\mathcal{L}_{\mathrm{Bi}}(\mathbf{s}, \mathbf{t}) = \frac{1}{2}\big[\mathcal{L}_{\{\mathbf{\Theta}, \mathbf{\Phi}\}}(\mathbf{s}, \mathbf{t}) + \mathcal{L}_{\{\mathbf{\Phi}, \mathbf{\Theta}\}}(\mathbf{t}, \mathbf{s})\big], \tag{2}$$

where both information flows are weighted equally. On a practical note, we find that it is important to ensure that parameters are shared appropriately when mirroring the terms, and that the variance in the gradient estimator is controlled effectively. Please see Appendices B to D for further details.

## 3.3 Learning from Partial Observations

In practice, prohibitive costs on multimodal data collection and curation imply that observations can frequently be partial, i.e., have missing modalities. One of the main benefits of the method introduced here is its natural extension to the case of partial observations on account of its semi-supervised underpinnings. Consider, without loss of generality, the case where we observe modality $\mathbf{s}$, but not its pair $\mathbf{t}$. Recalling the autoregressive generative model $p(\mathbf{s}, \mathbf{z}, \mathbf{t}) = p(\mathbf{s} \mid \mathbf{z})p(\mathbf{z} \mid \mathbf{t})p(\mathbf{t})$ we can derive a lower bound on the log-evidence

$$\log p_{\theta, \psi}(\mathbf{s}) = \log \int p_\theta(\mathbf{s} \mid \mathbf{z})p_\psi(\mathbf{z} \mid \mathbf{t})p(\mathbf{t}) \, d\mathbf{z} \, d\mathbf{t} \geq \mathbb{E}_{q_\phi(\mathbf{z}\mid\mathbf{s})}\left[\log \frac{p_\theta(\mathbf{s} \mid \mathbf{z}) \int p_\psi(\mathbf{z} \mid \mathbf{t})p(\mathbf{t}) \, d\mathbf{t}}{q_\phi(\mathbf{z} \mid \mathbf{s})}\right]. \tag{3}$$

Estimating the integral $p(\mathbf{z}) = \int p(\mathbf{z} \mid \mathbf{t})p(\mathbf{t}) \, d\mathbf{t}$ highlights another conceptual difference between a (semi-)supervised setting and a multimodal one. When $\mathbf{t}$ is seen as a label, this typically implies that one could possibly compute the integral *exactly* by explicit marginalisation over its support, or at the very least, construct a reasonable estimate through simple Monte-Carlo integration. In Joy et al. (2021), the authors extend the latter approach through importance sampling with the "inner" encoder $q(\mathbf{t} \mid \mathbf{z})$, to construct a looser lower bound to (3).

In the multimodal setting however, this poses serious difficulties as the domain of the variable $\mathbf{t}$ is not simple categorical labels, but rather complex continuous-valued data. This rules out exact marginalisation, and renders further importance-sampling practically infeasible on account of the quality of samples one can expect from the encoder $q(\mathbf{t} \mid \mathbf{z})$ which itself is being learnt from

data. To overcome this issue and to ensure a flexible alternative, we adopt an approach inspired by the VampPrior (Tomczak & Welling, 2018). Noting that our formulation includes a conditional prior $p_\psi(\mathbf{z} \mid \mathbf{t})$, we introduce learnable pseudo-samples $\lambda^{\mathbf{t}} = \{\mathbf{u}_i^{\mathbf{t}}\}_{i=1}^N$ to estimate the prior as $p_{\lambda^{\mathbf{t}}}(\mathbf{z}) = \frac{1}{N} \sum_{i=1}^N p_\psi(\mathbf{z} \mid \mathbf{u}_i^{\mathbf{t}})$. Our objective for when $\mathbf{t}$ is unobserved is thus

$$\mathcal{L}(\mathbf{s}) = \mathbb{E}_{q_\phi(\mathbf{z}|\mathbf{s})} \left[ \log \frac{p_\theta(\mathbf{s} \mid \mathbf{z}) p_{\lambda^{\mathbf{t}}}(\mathbf{z})}{q_\phi(\mathbf{z} \mid \mathbf{s})} \right] = \mathbb{E}_{q_\phi(\mathbf{z}|\mathbf{s})} \left[ \log \frac{p_\theta(\mathbf{s} \mid \mathbf{z})}{q_\phi(\mathbf{z} \mid \mathbf{s})} + \log \frac{1}{N} \sum_{i=1}^N p_\psi(\mathbf{z} \mid \mathbf{u}_i^{\mathbf{t}}) \right], \quad (4)$$

where the equivalent objective for when $\mathbf{s}$ is missing can be derived in a similar way. For a dataset $\mathcal{D}$ containing partial observations the overall objective (to maximise) becomes

$$\sum_{\mathbf{s},\mathbf{t} \in \mathcal{D}} \log p_{\theta,\psi}(\mathbf{s}, \mathbf{t}) \geq \sum_{\mathbf{s} \in \mathcal{D}_\mathbf{s}} \mathcal{L}(\mathbf{s}) + \sum_{\mathbf{t} \in \mathcal{D}_\mathbf{t}} \mathcal{L}(\mathbf{t}) + \sum_{\mathbf{s},\mathbf{t} \in \mathcal{D}_{\mathbf{s},\mathbf{t}}} \mathcal{L}_{\mathrm{Bi}}(\mathbf{s}, \mathbf{t}), \quad (5)$$

This treatment of unobserved data distinguishes our approach from alternatives such as that of Shi et al. (2019), where model updates for missing modalities are infeasible. Whilst there is the possibility to perform multimodal learning in the weakly supervised case as introduced by Wu & Goodman (2018), their approach directly affects the posterior distribution, whereas ours only affects the regularization of the embedding during training. At test time, Wu & Goodman (2018) will produce different embeddings depending on whether all modalities are present, which is typically at odds with the concept of placing the embeddings of related modalities in the same region of the latent space. Our approach does not suffer from this issue as the posterior remains unchanged regardless of whether the other modality is present or not.

**Learning with MEME** Given the overall objective in (5), we train MEME through maximum-likelihood estimation of the objective over a dataset $\mathcal{D}$. Each observation from the dataset is optimised using the relevant term in the right-hand side of (5), through the use of standard stocastic gradient descent methods. Note that training the objective involves learning *all* the (neural network) parameters $(\theta, \psi, \phi, \varphi)$ in the fully-observed, bi-directional case. When training with a partial observation, say just $\mathbf{s}$, all parameters except the relevant likelihood parameter $\varphi$ (for $q_\varphi(\mathbf{t} \mid \mathbf{z})$) are learnt. Note that the encoding for data in the domain of $\mathbf{t}$ is still computed through the learnable pseudo-samples $\lambda^{\mathbf{t}}$. This is reversed when training on an observation with just $\mathbf{t}$.

**Generalisation beyond two modalities** We confine our attention here to the bi-modal case for two important reasons. Firstly, the number of modalities one typically encounters in the multimodal setting is fairly small to begin with. This is often a consequence of its motivation from embodied perception, where one is restricted by the relatively small number of senses available (e.g. sight, sound, proprioception). Furthermore, the vast majority of prior work on multimodal VAEs only really consider the bimodal setting (cf. § 2). Secondly, it is quite straightforward to extend MEME to settings beyond the bimodal case, by simply incorporating existing explicit combinations (e.g. mixtures or products) *on top of* the implicit combination discussed here, we provide further explanation in Appendix E. Our focus in this work lies in exploring and analysing the utility of implicit combination in the multimodal setting, and our formulation and experiments reflect this focus.

## 4 EXPERIMENTS

### 4.1 LEARNING FROM PARTIALLY OBSERVED DATA

In this section, we evaluate the performance of MEME following standard multimodal VAE metrics as proposed in Shi et al. (2019). Since our model benefits from its implicit latent regularisation and is able to learn from partially-observed data, here we evaluate MEME's performance when different proportions of data are missing in either or both modalities during training. The two metrics used are *cross coherence* to evaluate the semantic consistency in the reconstructions, as well as *latent accuracy* in a classification task to quantitatively evaluate the representation learnt in the latent space. We demonstrate our results on two datasets, namely an image $\leftrightarrow$ image dataset MNIST-SVHN (LeCun et al., 2010; Netzer et al., 2011), which is commonly used to evaluate multimodal VAEs (Shi et al., 2019; Shi et al., 2021; Sutter et al., 2020; 2021); as well as the more challenging, but less common, image $\leftrightarrow$ caption dataset CUB (Welinder et al., 2010).

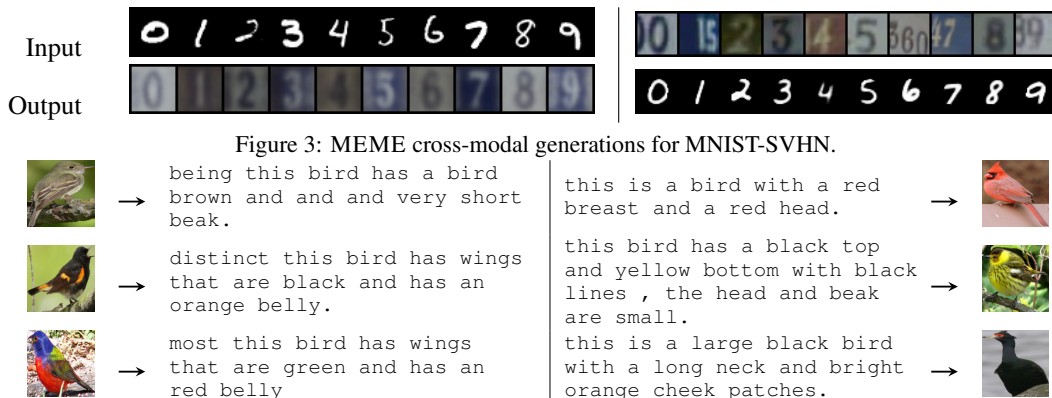

Input

Output

Figure 3: MEME cross-modal generations for MNIST-SVHN.

being this bird has a bird brown and and and very short beak.

this is a bird with a red breast and a red head.

distinct this bird has wings that are black and has an orange belly.

this bird has a black top and yellow bottom with black lines , the head and beak are small.

most this bird has wings that are green and has an red belly

this is a large black bird with a long neck and bright orange cheek patches.

Figure 4: MEME cross-modal generations for CUB.

Following standard approaches, we represented image likelihoods using Laplace distributions, and a categorical distribution for caption data. The latent variables are parameterised by Gaussian distributions. In line with previous research (Shi et al., 2019; Massiceti et al., 2018), simple convolutional architectures were used for both MNIST-SVHN and for CUB images *and* captions. For details on training and exact architectures see Appendix K; we also provide tabularised results in Appendix H.

**Cross Coherence**   Here, we focus mainly on the model's ability to reconstruct one modality, say, $t$, given another modality, $s$, as input, while preserving the conceptual commonality between the two. In keeping with Shi et al. (2019), we report the cross coherence score on MNIST-SVHN as the percentage of matching digit predictions of the input and output modality obtained from a pre-trained classifier. On CUB we perform canonical correlation analysis (CCA) on input-output pairs of cross generation to measure the correlation between these samples. For more details on the computation of CCA values we refer to Appendix G.

In Figure 5 we plot cross coherence for MNIST-SVHN and display correlation results for CUB in Figure 6, across different partial-observation schemes. The $x$-axis represents the proportion of data that is paired, while the subscript to the method (see legends) indicates the modality that is presented. For instance, MEME_MNIST with $f = 0.25$ indicates that only 25% of samples are paired, and the other 75% only contain MNIST digits, and MEME_SPLIT with $f = 0.25$ indicates that the 75% contains a mix of MNIST and SVHN samples that are unpaired and never observed together, i.e we alternate depending on the iteration, the remaining 25% contain paired samples. We provide qualitative results in Figure 3 and Figure 4.

We can see that our model is able to obtain higher coherence scores than the baselines including MVAE (Wu & Goodman, 2018) and MMVAE (Shi et al., 2019) in the fully observed case, $f = 1.0$, as well as in the case of partial observations, $f < 1.0$. This holds true for both MNIST-SVHN and CUB[2]. It is worth pointing out that the coherence between SVHN and MNIST is similar for both partially observing MNIST or SVHN, i.e. generating MNIST digits from SVHN is more robust to which modalities are observed during training (Figure 5 Right). However, when generating SVHN from MNIST, this is not the case, as when partially observing MNIST during training the model struggles to generate appropriate SVHN digits. This behaviour is somewhat expected since the information needed to generate an MNIST digit is typically subsumed within an SVHN digit (e.g. there is little style information associated with MNIST), making generation from SVHN to MNIST easier, and from MNIST to SVHN more difficult. Moreover, we also hypothesise that observing MNIST during training provides greater clustering in the latent space, which seems to aid cross generating SVHN digits. We provide additional t-SNE plots in Appendix H.3 to justify this claim.

For CUB we can see in Figure 6 that MEME consistently obtains higher correlations than MVAE across all supervision rates, and higher than MMVAE in the fully supervised case. Generally, cross-generating images yields higher correlation values, possibly due to the difficulty in generating semantically meaningful text with relatively simplistic convolutional architectures. We would like to highlight that partially observing captions typically leads to poorer performance when cross-

---

[2]We note that some of the reported results of MMVAE in our experiments do not match those seen in the original paper, please visit Appendix I for more information.

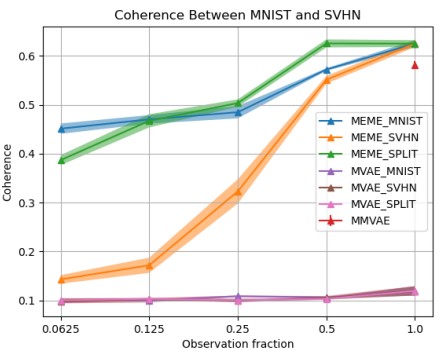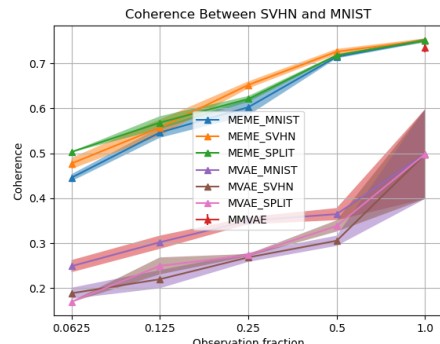

Figure 5: Coherence between MNIST and SVHN (Left) and SVHN and MNIST (Right). Shaded area indicates one-standard deviation of runs with different seeds.

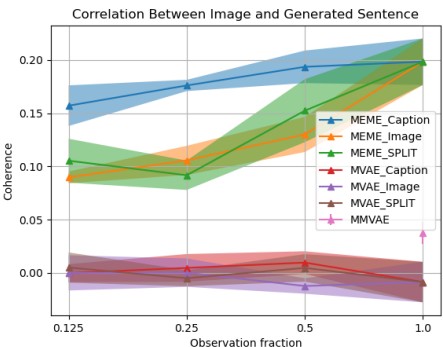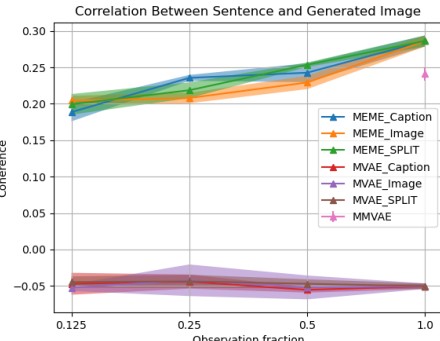

Figure 6: Correlation between Image and Sentence (Left) and Sentence and Image (Right). Shaded area indicates one-standard deviation of runs with different seeds.

generating captions. We hypothesise that is due to the difficulty in generating the captions and the fact there is a limited amount of captions data in this setting.

**Latent Accuracy**    To gauge the quality of the learnt representations we follow previous work (Higgins et al., 2017; Kim & Mnih, 2018; Shi et al., 2019; Sutter et al., 2021) and fit a linear classifier that predicts the input digit from the latent samples. The accuracy of predicting the input digit using this classifier indicates how well the latents can be separated in a linear manner.

In Figure 7, we plot the latent accuracy on MNIST and SVHN against the fraction of observation. We can see that MEME outperforms MVAE on both MNIST and SVHN under the fully-observed scheme (i.e. when observation fractions is $1.0$). We can also notice that the latent accuracy of MVAE is rather lopsided, with the performance on MNIST to be as high as $0.88$ when only 1/16 of the data is observed, while SVHN predictions remain almost random even when all data are used; this indicates that MVAE relies heavily on MNIST to extract digit information. On the other hand, MEME's latent accuracy observes a steady increase as observation fractions grow in both modalities. It is worth noting that both models performs better on MNIST than SVHN in general—this is unsurprising as it is easier to disentangle digit information from MNIST, however our experiments here show that MEME does not completely disregard the digits in SVHN like MVAE does, resulting in more balanced learned representations. It is also interesting to see that MVAE obtains a higher latent accuracy than MEME for low supervision rates. This is due to MVAE learning to construct representations for each modality in a completely separate sub-space in the latent space, we provide a t-SNE plot to demonstrate this in Appendix H.1.

**Ablation Studies**    To study the effect of modelling and data choices on performance, we perform two ablation studies: one varying the number of pseudo-samples for the prior, and the other evaluating how well the model leverages partially observed data over fully observed data. We find that performance degrades, as expected, with fewer pseudo-samples, and that the model trained with additional partially observed data does indeed improve. See Appendix J for details.

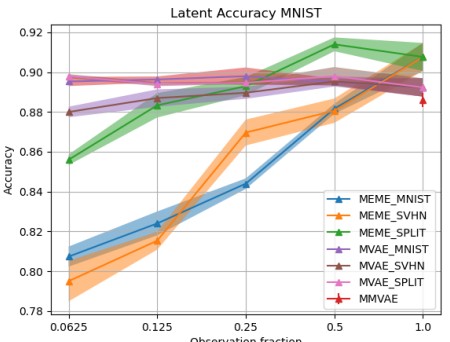 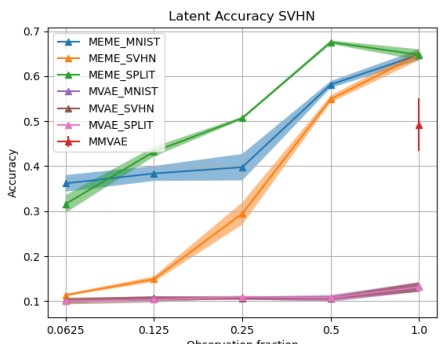

Figure 7: Latent accuracies for MNIST and SVHN (Left) and SVHN and MNIST (Right). Shaded area indicates one-standard deviation of runs with different seeds.

## 4.2 EVALUATING RELATEDNESS

Now that we have established that the representation learned by MEME contains rich class information from the inputs, we also wish to analyse the relationship between the encodings of different modalities by studying their "relatedness", i.e. semantic similarity. The probabilistic nature of the learned representations suggests the use of probability distance functions as a measure of relatedness, where a low distance implies closely related representations and vice versa.

In the following experiments we use the 2-Wasserstein distance, $\mathcal{W}_2$, a probability metric with a closed-form expression for Gaussian distributions (see Appendix F for more details). Specifically, we compute $d_{ij} = \mathcal{W}_2(q(\mathbf{z}|\mathbf{s}_i) \| q(\mathbf{z}|\mathbf{t}_j))$, where $q(\mathbf{z}|\mathbf{s}_i)$ and $q(\mathbf{z}|\mathbf{t}_j)$ are the individual encoders, for all combination of pairs $\{\mathbf{s}_i, \mathbf{t}_j\}$ in the mini-batch, i.e $\{\mathbf{s}_i, \mathbf{t}_j\}$, for $i, j \in \{1 \ldots, M\}$ where $M$ is the number of elements in the mini-batch.

**General Relatedness** In this experiment we wish to highlight the disparity in measured relatedness between paired vs. unpaired multimodal data. To do so, we plot $d_{ij}$ on a histogram and color-code the histogram by whether the corresponding data pair $\{\mathbf{s}_i, \mathbf{t}_j\}$ shows the same concept, e.g. same digit for MNIST-SVHN and same image-caption pair for CUB. Ideally, we should observe smaller distances between encoding distributions for data pairs that are related, and larger for ones that are not.

To investigate this, we plot $d_{ij}$ on a histogram for every mini-batch; ideally we should see higher densities at closer distances for points that are paired, and higher densities at further distances for unpaired points. In Figure 8, we see that MEME (left) does in fact yields higher mass at lower

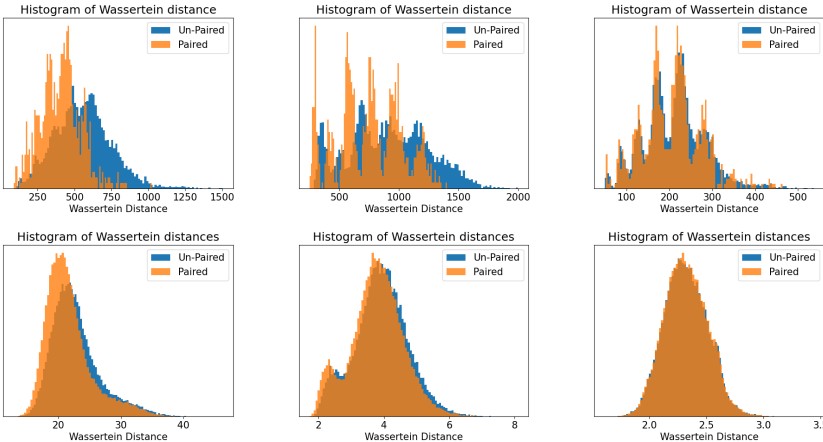

Figure 8: Histograms of Wassertein distance for SVHN and MNIST (Top) and CUB (Bottom): MEME (Left), MMVAE (middle) and MVAE (Right). Blue indicates *unpaired* samples and orange *paired* samples. We expect to see high densities of blue at further distances and visa-versa for orange.

distance values for paired multimodal samples (orange) than it does for unpaired ones (blue). This effect is not so pronounced in MMVAE and not present at all in MVAE. This demonstrates MEME's capability of capturing relatedness between multimodal samples in its latent space, and the quality of its representation.

**Class-contextual Relatedness**    To offer more insights on the relatedness of representations within classes, we construct a distance matrix $\mathbf{K} \in \mathbb{R}^{10 \times 10}$ for the MNIST-SVHN dataset, where each element $\mathbf{K}_{i,j}$ corresponds to the average $\mathcal{W}_2$ distance between encoding distributions of class $i$ of MNIST and $j$ of SVHN. A perfect distance matrix will consist of a diagonal of all zeros and positive values in the off-diagonal.

See the class distance matrix in Figure 9 (top row), generated with models trained on fully observed multimodal data. It is clear that our model (left) produces much lower distances on the diagonal, i.e. when input classes for the two modalities are the same, and higher distances off diagonal where input classes are different. A clear, lower-valued diagonal can also be observed for MMVAE (middle), however it is less distinct compared to MEME, since some of the mismatched pairs also obtains smaller values. The distance matrix for MVAE (right), on the other hand, does not display a diagonal at all, reflecting poor ability to identify relatedness or extract class information through the latent.

To closely examine which digits are considered similar by the model, we construct dendrograms to visualise the hierarchical clustering of digits by relatedness, as seen in Figure 9 (bottom row). We see that our model (left) is able to obtain a clustering of conceptually similar digits. In particular, digits with smoother writing profile such as 3, 5, 8, along with 6 and 9 are clustered together (right hand side of dendrogram), and the digits with sharp angles, such as 4 and 7 are clustered together. The same trend is not observed for MMVAE nor MVAE. It is also important to note the height of each bin, where higher values indicate greater distance between clusters. Generally the clusters obtained in MEME are further separated for MMVAE, demonstrating more distinct clustering across classes.

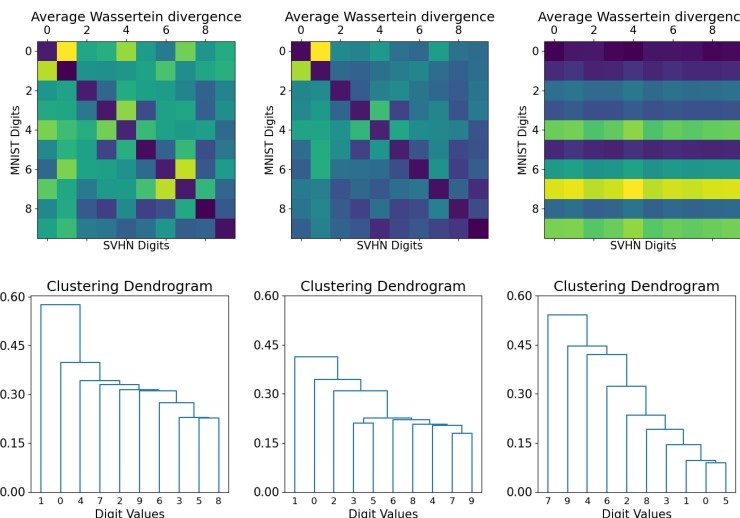

Figure 9: Distance matrices for KL divergence between classes for SVHN and MNIST (Top) and dendrogram (Bottom) for: Ours (Left), MMVAE (middle) and MVAE (Right).

# 5    DISCUSSION

Here we have presented a method which faithfully deals with partially observed modalities in VAEs. Through leveraging recent advances in semi-supervised VAEs, we construct a model which is amenable to multi-modal learning when modalities are partially observed. Specifically, our method employs mutual supervision by treating the uni-modal encoders individually and minimizing a KL between them to ensure embeddings for are pertinent to one another. This approach enables us to successfully learn a model when either of the modalites are partially observed. Furthermore, our model is able to naturally extract an indication of relatedness between modalities. We demonstrate our approach on the MNIST-SVHN and CUB datasets, where training is performed on a variety of different observations rates.

**Ethics Statement**   We believe there are no inherent ethical concerns within this work, as all datasets and motivations do not include or concern humans. As with every technological advancement there is always the potential for miss-use, for this work though, we can not see a situation where this method may act adversarial to society. In fact, we believe that multi-modal representation learning in general holds many benefits, for instance in language translation which removes the need to translate to a base language (normally English) first.

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

## A   DERIVATION OF THE OBJECTIVE

The variational lower bound for the case when $\mathbf{s}$ and $\mathbf{t}$ are both observed, following the notation in Figure 2, derives as:

$$\log p_{\theta,\psi}(\mathbf{t},\mathbf{s}) = \log \int_{\mathbf{z}} p_{\theta,\psi}(\mathbf{t},\mathbf{s},\mathbf{z})d\mathbf{z}$$

$$\geq \int_{\mathbf{z}} \log \frac{p_{\theta,\psi}(\mathbf{s},\mathbf{t},\mathbf{z})}{q_{\phi,\varphi}(\mathbf{z}|\mathbf{t},\mathbf{s})} q_{\phi,\varphi}(\mathbf{z}|\mathbf{t},\mathbf{s})d\mathbf{z}$$

Following Joy et al. (2021), assuming $\mathbf{s} \perp\!\!\!\perp \mathbf{t}|\mathbf{z}$ and applying Bayes rule we have

$$q_{\phi,\varphi}(\mathbf{z}|\mathbf{t},\mathbf{s}) = \frac{q_\phi(\mathbf{z}|\mathbf{s})q_\varphi(\mathbf{t}|\mathbf{z})}{q_{\phi,\varphi}(\mathbf{t}|\mathbf{s})}, \quad \text{where } q_{\phi,\varphi}(\mathbf{t}|\mathbf{s}) = \int q_\phi(\mathbf{z}|\mathbf{s})q_\varphi(\mathbf{t}|\mathbf{z})d\mathbf{z}$$

which can be substituted into the lower bound to obtain

$$\log p_{\theta,\psi}(\mathbf{t},\mathbf{s}) \geq \int_{\mathbf{z}} \log \frac{p_{\theta,\psi}(\mathbf{s},\mathbf{t},\mathbf{z})q_{\phi,\varphi}(\mathbf{t}|\mathbf{s})}{q_\phi(\mathbf{z}|\mathbf{s})q_\varphi(\mathbf{t}|\mathbf{z})} \frac{q_\phi(\mathbf{z}|\mathbf{s})q_\varphi(\mathbf{t}|\mathbf{z})}{q_{\phi,\varphi}(\mathbf{t}|\mathbf{s})}d\mathbf{z}$$

$$= \mathbb{E}_{q_\phi(\mathbf{z}|\mathbf{s})}\left[ \frac{q_\varphi(\mathbf{t}|\mathbf{z})}{q_{\phi,\varphi}(\mathbf{t}|\mathbf{s})} \log \frac{p_\theta(\mathbf{s}|\mathbf{z})p_\psi(\mathbf{z}|\mathbf{t})}{q_\phi(\mathbf{z}|\mathbf{s})q_\varphi(\mathbf{t}|\mathbf{z})} \right] + \log q_{\phi,\varphi}(\mathbf{t}|\mathbf{s}) + \log p(\mathbf{t}). \quad (6)$$

## B   EFFICIENT GRADIENT ESTIMATION

Given the objective in (6), note that the first term is quite complex, and requires estimating a weight ratio that involves an additional integral for $q_{\varphi,\phi}(\mathbf{t} \mid \mathbf{s})$. This has a significant effect, as the naive Monte-Carlo estimator of

$$\nabla_{\phi,\varphi}\mathbb{E}_{q_\phi(\mathbf{z}|\mathbf{s})}\left[ \frac{q_\varphi(\mathbf{t} \mid \mathbf{z})}{q_{\varphi,\phi}(\mathbf{t} \mid \mathbf{s})} \log \frac{p_\theta(\mathbf{s} \mid \mathbf{z})p_\psi(\mathbf{z} \mid \mathbf{t})}{q_\phi(\mathbf{z} \mid \mathbf{s})q_\varphi(\mathbf{t} \mid \mathbf{z})} \right]$$

$$= \mathbb{E}_{p(\epsilon)}\left[ \left( \nabla_{\phi,\varphi} \frac{q_\varphi(\mathbf{t} \mid \mathbf{z})}{q_{\varphi,\phi}(\mathbf{t} \mid \mathbf{s})} \right) \log \frac{p_\theta(\mathbf{s} \mid \mathbf{z})p_\psi(\mathbf{z} \mid \mathbf{t})}{q_\phi(\mathbf{z} \mid \mathbf{s})q_\varphi(\mathbf{t} \mid \mathbf{z})} + \frac{q_\varphi(\mathbf{t} \mid \mathbf{z})}{q_{\varphi,\phi}(\mathbf{t} \mid \mathbf{s})} \nabla_{\phi,\varphi} \log \frac{p_\theta(\mathbf{s} \mid \mathbf{z})p_\psi(\mathbf{z} \mid \mathbf{t})}{q_\phi(\mathbf{z} \mid \mathbf{s})q_\varphi(\mathbf{t} \mid \mathbf{z})} \right] \quad (7)$$

can be very noisy, and prohibit learning effectively. To mitigate this, we note that the first term in (7) computes gradients for the encoder parameters $(\phi, \varphi)$ through a ratio of probabilities, whereas the second term does so through $\log$ probabilities. Numerically, the latter is a lot more stable to learn from than the former, and so we simply drop the first term in (7) by employing a `stop_gradient` on the ratio $\frac{q_\varphi(\mathbf{t}|\mathbf{z})}{q_{\varphi,\phi}(\mathbf{t}|\mathbf{s})}$. We further support this change with empirical results (cf. Figure 10) that show how badly the signal-to-noise ratio (SNR) is affected for the gradients with respect to the encoder parameters. We further note that Joy et al. (2021) perform a similar modification also motivated by an empirical study, but where they detach the sampled $\mathbf{z}$—we find that our simplification that detaches the weight itself works more stably and effectively.

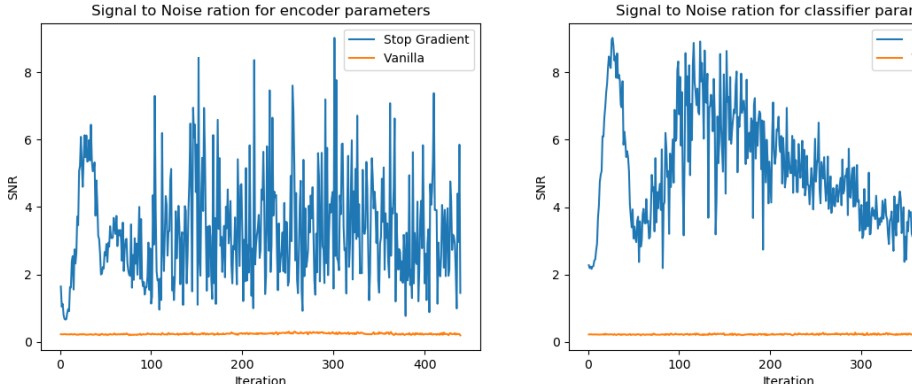

Figure 10: SNR for parameters ($\phi$, left) and ($\varphi$, right). The blue curve denotes the simplified estimator using stop gradient, and the orange curve indicates the full estimator in (7). Higher values leads to improved learning.

## C  WEIGHT SHARING

Another critical issue with naïvely training using (6), is that in certain situations $q_\varphi(\mathbf{t} \mid \mathbf{z})$ struggles to learn features (typically style) for $\mathbf{t}$, consequently making it difficult to generate realistic samples. This is due to the information entering the latent space only coming from $\mathbf{s}$, which contains all of the information needed to reconstruct $\mathbf{s}$, but does not necessarily contain the information needed to reconstruct a corresponding $\mathbf{t}$. Consequently, the term $p_\theta(\mathbf{s} \mid \mathbf{z})$ will learn appropriate features (like a standard VAE decoder), but the term $q_\varphi(\mathbf{t} \mid \mathbf{z})$ will fail to do so. In situations like this, where the information in $\mathbf{t}$ is *not* subsumed by the information in $\mathbf{s}$, there is no way for the model to know how to reconstruct a $\mathbf{t}$. Introducing weight sharing into the bidirectional objective (2) removes this issue, as there is equal opportunity for information from both modalities to enter the latent space, consequently enabling appropriate features to be learned in the decoders $p_\theta(\mathbf{s} \mid \mathbf{z})$ and $p_\varphi(\mathbf{t} \mid \mathbf{z})$, which subsequently allow cross generations to be performed.

Furthermore, we also observe that when training with (2) we are able to obtain much more balanced likelihoods Table 2. In this setting we train two models separately using (6) with $\mathbf{s}$ = MNIST and SVHN and then with $\mathbf{t}$ = SVHN and $\mathbf{s}$ = MNIST respectively. At test time, we then 'flip' the modalities and the corresponding networks, allowing us to obtain marginal likelihoods in each direction. Clearly we see that we only obtain reasonable marginal likelihoods in the direction for which we train. Training with the bidirectional objective completely removes this deficiency, as we now introduce a balance between the modalities.

Table 2: Marginal likelihoods.

| Test Direction | Train Direction | | |
| --- | --- | --- | --- |
| | s = M, t = S | s = S, t = M | Bi |
| s = M, t = S | $-14733.6$ | $-40249.9^{\text{flip}}$ | $-14761.3$ |
| s = S, t = M | $-428728.7^{\text{flip}}$ | $-11668.1$ | $-11355.4$ |

## D  REUSING APPROXIMATE POSTERIOR MC SAMPLE

When approximating $q_{\varphi,\phi}(\mathbf{t} \mid \mathbf{s})$ through MC sampling, we find that it is essential for numerical stability to include the sample from the approximate posterior. Before considering why, we must first outline the numerical implementation of $q_{\varphi,\phi}(\mathbf{t} \mid \mathbf{s})$, which for $K$ samples $\mathbf{z}_{1:K} \sim q_\phi(\mathbf{z} \mid \mathbf{s})$ is computed using the LogSumExp trick as:

$$\log q_{\varphi,\phi}(\mathbf{t} \mid \mathbf{s}) \approx \log \sum_{k=1}^{K} \exp \log q_\varphi(\mathbf{t}|\mathbf{z}_k), \tag{8}$$

where the ratio $\frac{q_\varphi(\mathbf{t}|\mathbf{z})}{q_{\varphi,\phi}(\mathbf{t}|\mathbf{s})}$ is computed as $\exp\{\log q_\varphi(\mathbf{t} \mid \mathbf{z}) - \log q_{\varphi,\phi}(\mathbf{t} \mid \mathbf{s})\}$. Given that the LogSumExp trick is defined as:

$$\log \sum_{n=1}^{N} \exp x_n = x^* + \log \sum_{n=1}^{N} \exp(x_n - x^*), \tag{9}$$

where $x^* = \max\{x_1, \ldots, x_N\}$. The ratio will be computed as

$$\frac{q_\varphi(\mathbf{t} \mid \mathbf{z})}{q_{\varphi,\phi}(\mathbf{t} \mid \mathbf{s})} = \exp\{\log q_\varphi(\mathbf{t} \mid \mathbf{z}) - \log q_\varphi(\mathbf{t}|\mathbf{z}^*) - \log \sum_{k=1}^{K} \exp[\log q_\varphi(\mathbf{t}|\mathbf{z}_k) - \log q_\varphi(\mathbf{t}|\mathbf{z}^*)]\},$$

$$\tag{10}$$

where $\mathbf{z}^* = \arg\max_{\mathbf{z}_{1:K}} \log q_\varphi(\mathbf{t}|\mathbf{z}_k)$. For numerical stability, we require that $\log q_\varphi(\mathbf{t} \mid \mathbf{z}) \not\gg \log q_\varphi(\mathbf{t}|\mathbf{z}^*)$, otherwise the computation may blow up when taking the exponent. To enforce this, we need to include the sample $\mathbf{z}$ into the LogSumExp function, doing so will cause the first two terms to either cancel if $\mathbf{z} = \mathbf{z}^*$ or yield a negative value, consequently leading to stable computation when taking the exponent.

# E    EXTENSION BEYOND THE BI-MODAL CASE

Here we offer further detail on how MEME can be extended beyond the bi-modal case, i.e. when the number of modalities $M > 2$. Note that the central thesis in MEME is that the evidence lower bound (ELBO) offers an implicit way to regularise different representations if viewed from the posterior-prior perspective, which can be used to build effective multimodal DGMs that are additionally applicable to partially-observed data. In MEME, we explore the utility of this implicit regularisation in the simplest possible manner to show that a direct application of this to the multi-modal setting would involve the case where $M = 2$.

The way to extend, say for $M = 3$, involves additionally employing an explicit combination for two modalities in the prior (instead of just 1). This additional combination could be something like a mixture or product, following from previous approaches. More formally, if we were to denote the implicit regularisation between posterior and prior as $R_i(.,.)$, and an explicit regularisation function $R_e(.,.)$, and the three modalities as $m_1, m_2$, and $m_3$, this would mean we would compute

$$\frac{1}{3} \left[ R_i(m_1, R_e(m_2, m_3)) + R_i(m_2, R_e(m_1, m_3)) + R_i(m_3, R_e(m_1, m_2)) \right], \tag{11}$$

assuming that $R_e$ was commutative, as is the case for products and mixtures. There are indeed more terms to compute now compared to $M = 2$, which only needs $R_i(m_1, m_2)$, but note that $R_i$ is still crucial—it does not diminish because we are additionally employing $R_e$.

As stated in prior work(Suzuki et al., 2016; Wu & Goodman, 2018; Shi et al., 2019), we follow the reasoning that the actual number of modalities, at least when considering embodied perception, is not likely to get much larger, so the increase in number of terms, while requiring more computation, is unlikely to become intractable. Note that prior work on multimodal VAEs also suffer when extending the number of modalities in terms of the number of paths information flows through.

We do not explore this setting empirically as our priary goal is to highlight the utility of this implicit regularisation for multi-modal DGMs, and its effectiveness at handling partially-observed data.

# F    CLOSED FORM EXPRESSION FOR WASSERTEIN DISTANCE BETWEEN TWO GAUSSIANS

The Wassertein-2 distance between two probability measures $\mu$ and $\nu$ on $\mathbb{R}^n$ is defined as

$$\mathcal{W}_2(\mu, \nu) := \inf \mathbb{E}(||X - Y||_2^2)^{\frac{1}{2}},$$

with $X \sim \mu$ and $Y \sim \nu$. Given $\mu = \mathcal{N}(m_1, \Sigma_1)$ and $\nu = \mathcal{N}(m_2, \Sigma_2)$, the 2-Wassertein is then given as

$$d^2 = ||m_1 + m_2||_2^2 + \mathrm{Tr}(\Sigma_1 + \Sigma_2 - 2(\Sigma_1^{\frac{1}{2}} \Sigma_2 \Sigma_1^{\frac{1}{2}})^{\frac{1}{2}}).$$

For a detailed proof please see (Givens & Shortt, 2002).

# G    CANONICAL CORRELATION ANALYSIS

Following Shi et al. (2019); Massiceti et al. (2018), we report cross-coherence scores for CUB using Canonical Correlation Analysis (CCA). Given paired observations $\mathbf{x}_1 \in \mathbb{R}_1^n$ and $\mathbf{x}_2 \in \mathbb{R}_2^n$, CCA learns projection weights $W_1^T \in \mathbb{R}^{n_1 \times k}$ and $W_2^T \in \mathbb{R}^{n_2 \times k}$ which minimise the correlation between the projections $W_1^T \mathbf{x}_1$ and $W_2^T \mathbf{x}_2$. The correlations between a data pair $\{\tilde{\mathbf{x}}_1, \tilde{\mathbf{x}}_2\}$ can thus be calculated as

$$\mathrm{corr}(\tilde{\mathbf{x}}_1, \tilde{\mathbf{x}}_2) = \frac{\phi(\tilde{\mathbf{x}}_1)^T \phi(\tilde{\mathbf{x}}_2)}{||\phi(\tilde{\mathbf{x}}_1)||_2 ||\phi(\tilde{\mathbf{x}}_2)||_2} \tag{12}$$

where $\phi(\mathbf{x}_n) = W_n^T \tilde{\mathbf{x}}_n - \mathrm{avg}(W_n^T \tilde{\mathbf{x}}_n)$.

Following Shi et al. (2019), we use feature extractors to pre-process the data. Specifically, features for image data are generated from an off-the-shelf ResNet-101 network. For text data, we first fit a FastText model on all sentences, resulting in a 300-$d$ projection for each word Bojanowski et al. (2017), the representation is then computed as the average over the words in the sentence.

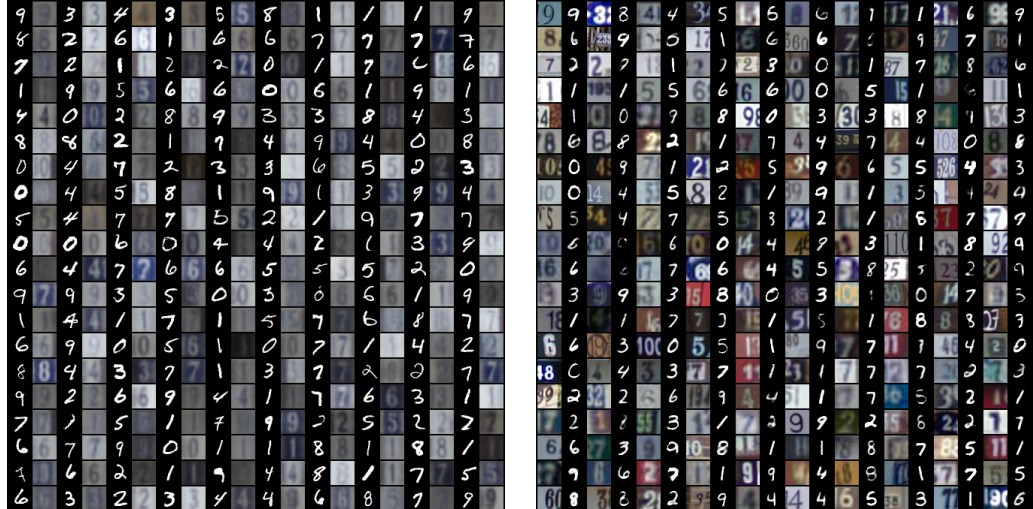

Figure 11: MNIST → SVHN (Left) and SVHN → MNIST (Right), for the fully observed case.

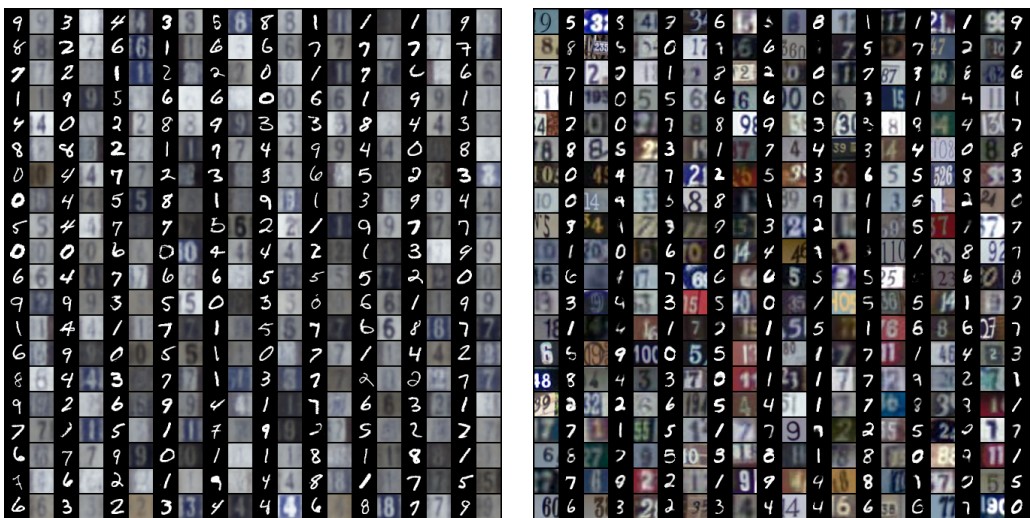

Figure 12: MNIST → SVHN (Left) and SVHN → MNIST (Right), when SVHN is observed 50% of the time.

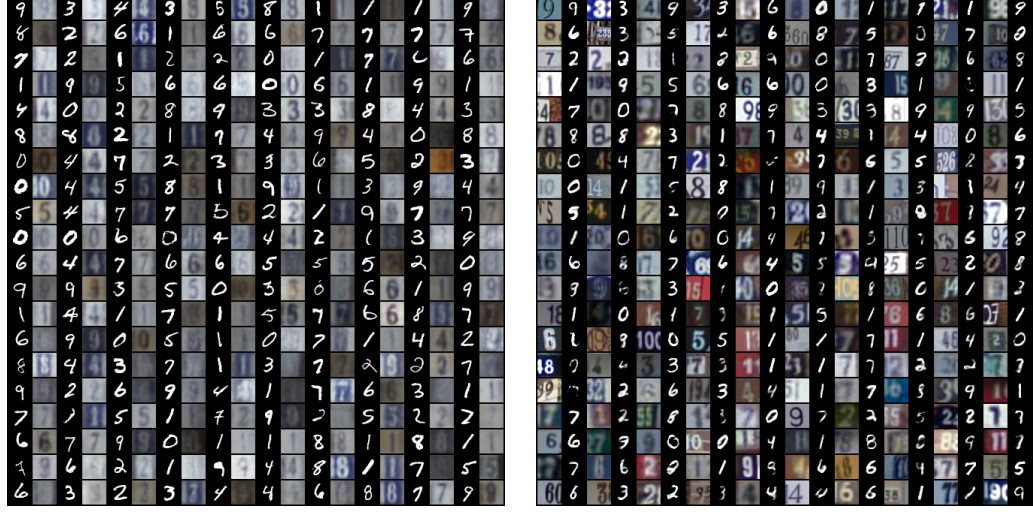

Figure 13: MNIST → SVHN (Left) and SVHN → MNIST (Right), when MNIST is observed 50% of the time.

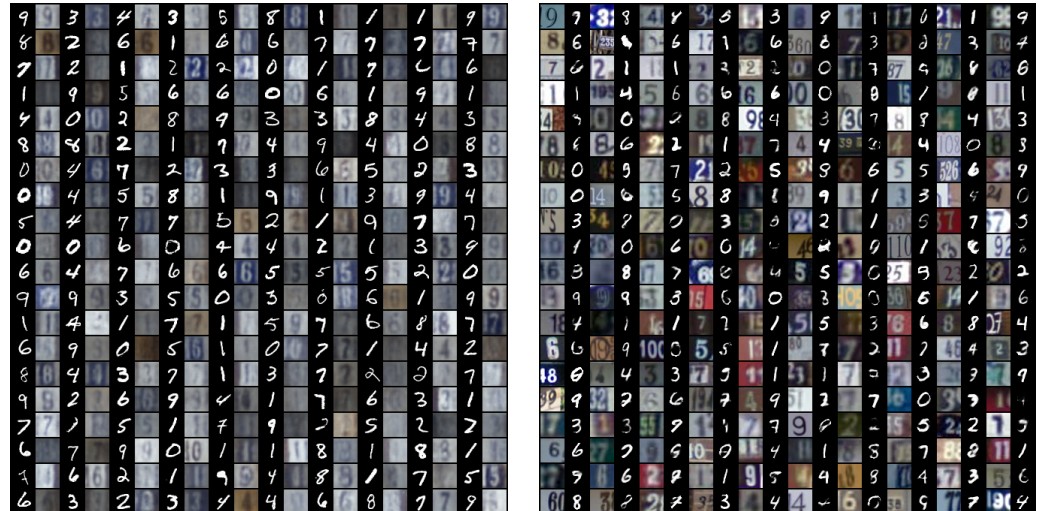

Figure 14: MNIST → SVHN (Left) and SVHN → MNIST (Right), when SVHN is observed 25% of the time.

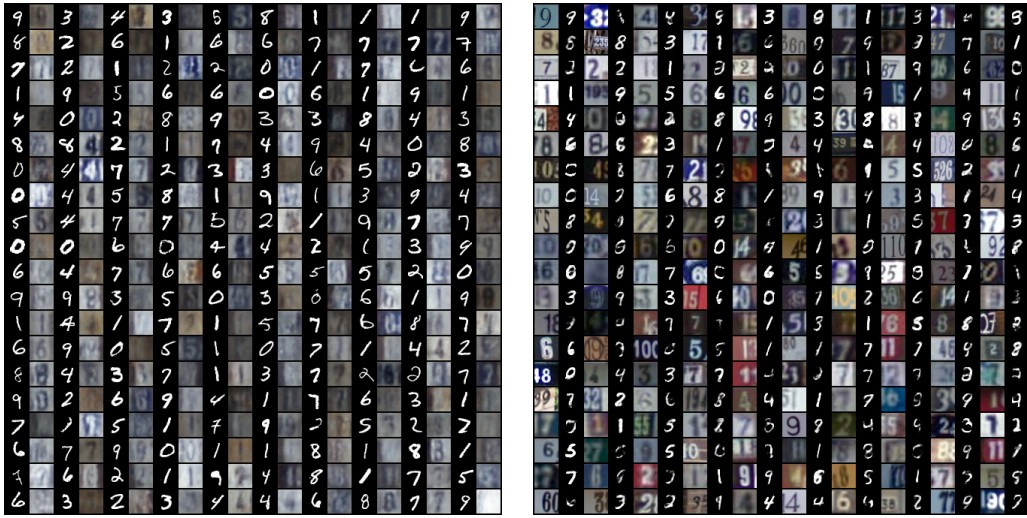

Figure 15: MNIST → SVHN (Left) and SVHN → MNIST (Right), when MNIST is observed 25% of the time.

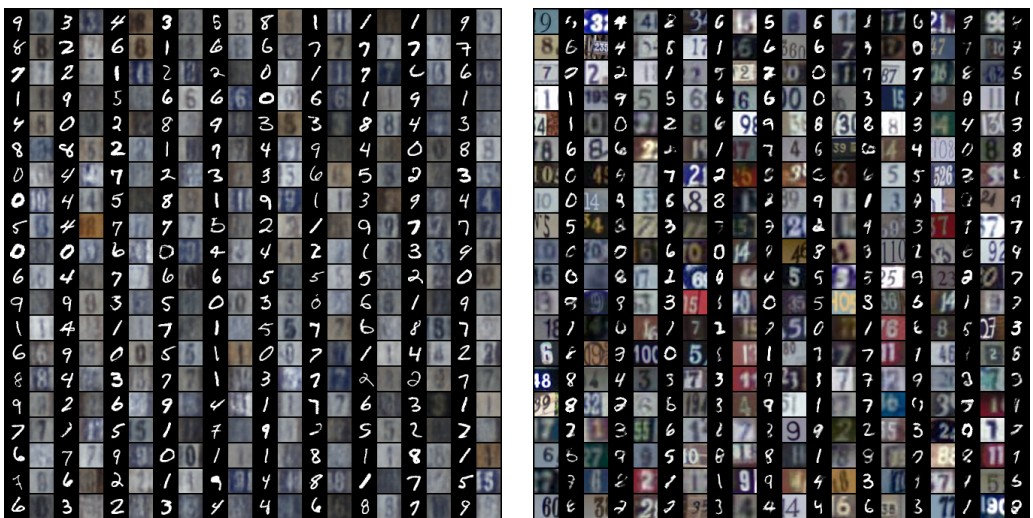

Figure 16: MNIST → SVHN (Left) and SVHN → MNIST (Right), when SVHN is observed 12.5% of the time.

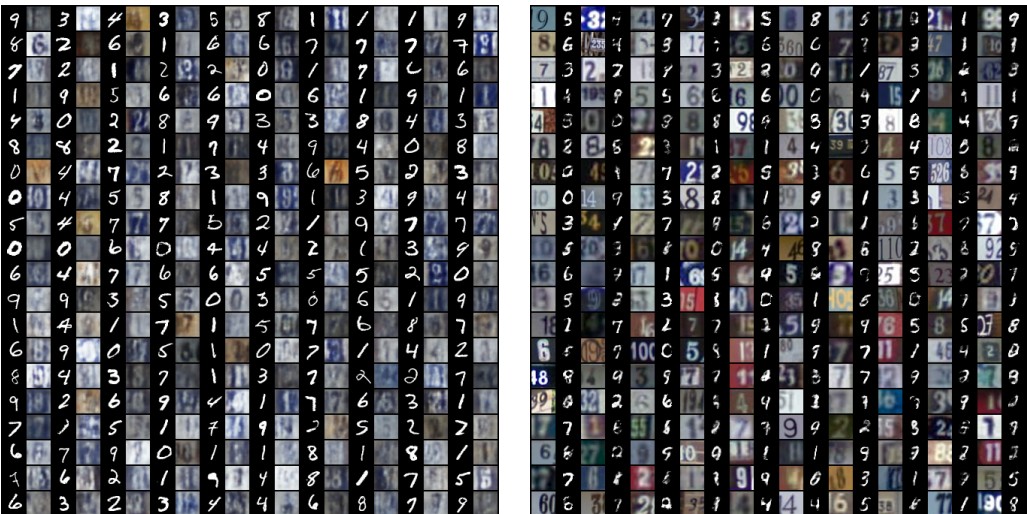

Figure 17: MNIST → SVHN (Left) and SVHN → MNIST (Right), when MNIST is observed 12.5% of the time.

Table 3: Coherence Scores for MNIST → SVHN (Top) and for SVHN → MNIST (Bottom). Subscript indicates which modality is always present during training, $f$ indicates the percentage of matched samples. Higher is better.

| | MNIST → SVHN | | | | |
|---|---|---|---|---|---|
| Model | $f = 1.0$ | $f = 0.5$ | $f = 0.25$ | $f = 0.125$ | $f = 0.0625$ |
| MEME$_{SVHN}$ | **0.625 ± 0.007** | **0.551 ± 0.008** | **0.323 ± 0.025** | **0.172 ± 0.016** | **0.143 ± 0.009** |
| MMVAE$_{SVHN}$ | 0.581 ± 0.008 | - | - | - | - |
| MVAE$_{SVHN}$ | 0.123 ± 0.003 | 0.110 ± 0.014 | 0.112 ± 0.005 | 0.105 ± 0.005 | 0.105 ± 0.006 |
| MEME$_{MNIST}$ | **0.625 ± 0.007** | **0.572 ± 0.003** | **0.485 ± 0.013** | **0.470 ± 0.009** | **0.451 ± 0.011** |
| MMVAE$_{MNIST}$ | 0.581 ± 0.008 | - | - | - | - |
| MVAE$_{MNIST}$ | 0.123 ± 0.003 | 0.111 ± 0.007 | 0.112 ± 0.013 | 0.116 ± 0.012 | 0.116 ± 0.005 |
| MEME$_{SPLIT}$ | **0.625 ± 0.007** | 0.625 ± 0.008 | 0.503 ± 0.008 | 0.467 ± 0.013 | 0.387 ± 0.010 |
| MVAE$_{SPLIT}$ | 0.123 ± 0.003 | 0.108 ± 0.005 | 0.101 ± 0.005 | 0.101 ± 0.001 | 0.101 ± 0.002 |

| | SVHN → MNIST | | | | |
|---|---|---|---|---|---|
| Model | $f = 1.0$ | $f = 0.5$ | $f = 0.25$ | $f = 0.125$ | $f = 0.0625$ |
| MEME$_{SVHN}$ | **0.752 ± 0.004** | **0.726 ± 0.006** | **0.652 ± 0.008** | **0.557 ± 0.018** | **0.477 ± 0.012** |
| MMVAE$_{SVHN}$ | 0.735 ± 0.010 | - | - | - | - |
| MVAE$_{SVHN}$ | 0.498 ± 0.100 | 0.305 ± 0.011 | 0.268 ± 0.010 | 0.220 ± 0.020 | 0.188 ± 0.012 |
| MEME$_{MNIST}$ | **0.752 ± 0.004** | **0.715 ± 0.003** | **0.603 ± 0.018** | **0.546 ± 0.012** | **0.446 ± 0.008** |
| MMVAE$_{MNIST}$ | 0.735 ± 0.010 | - | - | - | - |
| MVAE$_{MNIST}$ | 0.498 ± 0.100 | 0.365 ± 0.014 | 0.350 ± 0.008 | 0.302 ± 0.015 | 0.249 ± 0.014 |
| MEME$_{SPLIT}$ | **0.752 ± 0.004** | 0.718 ± 0.002 | 0.621 ± 0.007 | 0.568 ± 0.014 | 0.503 ± 0.001 |
| MVAE$_{SPLIT}$ | 0.498 ± 0.100 | 0.338 ± 0.013 | 0.273 ± 0.003 | 0.249 ± 0.019 | 0.169 ± 0.001 |

# H ADDITIONAL RESULTS

## H.1 MVAE LATENT ACCURACIES

The superior accuracy in latent accuracy when classifying MNIST from MVAE is due to a complete failure to construct a joint representation, which is evidenced in its failure to perform cross-generation. Failure to construct joint representations aids latent classification, as the encoders just learn to construct representations for single modalities, this then provides more flexibility and hence better classification. In Figure 19, we further provide a t-SNE plot to demonstrate that MVAE places representations for MNIST modality in completely different parts of the latent space to SVHN. Here

Fully observed.

| | | | |
|---|---|---|---|
|  → | this is a white bird with a wings and and black beak. | a grey bird with darker brown mixed in and a short brown beak. → |  |
|  → | a small brown bird with a white belly. | yellow bird with a black and white wings with a black beak. → |  |

Captions observed 50% of the time.

| | | | |
|---|---|---|---|
|  → | than a particular a wings a with and wings has on yellow. | the bird has two large , grey wingbars , and orange feet. → |  |
|  → | below a small has has that bill yellow and. | yellow bird with a black and white wings with a black beak. → |  |

Images observed 50% of the time.

| | | | |
|---|---|---|---|
|  → | blacks this bird has has that are that and has a yellow belly. | tiny brown bird with white breast and a short stubby bill. → |  |
|  → | bird this bird has red with bird with a a and a pointy. | this is a yellow bird with a black crown on its head. → |  |

Captions observed 25% of the time.

| | | | |
|---|---|---|---|
|  → | throat a is is yellow with yellow gray chest has medium belly belly throat. | this is a puffy bird with a bright yellow chest with white streaks along the feathers. → |  |
|  → | bird green small has crown as wing crown white white abdomen and crown and and. | this bird has a small bill with a black head and wings but white body. → |  |

Images observed 25% of the time.

| | | | |
|---|---|---|---|
|  → | crest this small with looking with a brown with a and face body. | the bird had a large white breast in <exc> to its head size. → |  |
|  → | rotund white bill large bird , brown black back mostly with with breast flying. | this bird has a black belly , breast and head , gray and white wings , and red tarsus and feet. → |  |

Captions observed 12.5% of the time.

| | | | |
|---|---|---|---|
|  → | a the bird wings a is and nape the and , crown , rectrices grey , , beak its and , a tail. | white belly with a brown body and a very short , small brown beak. → |  |
|  → | red this bird skinny black black crown red feet beak downwards short and a. | a bird with a short , rounded beak which ends in a point , stark white eyes , and white throat. → |  |

Images observed 12.5% of the time.

| | | | |
|---|---|---|---|
|  → | an a is colorful bird , short with and black and crown small short orange light small light over. | small bird with a long beak and blue wing feathers with brown body. → |  |
|  → | a this is white a all is with flat beak and black a , 's a curved for light has a body is. | this is a large black bird with a long neck and bright orange cheek patches. → |  |

Figure 18: MEME cross-modal generations for CUB.

we can see that representations for each modality are completely separated, meaning that there is no shared representation. Furthermore, MNIST is well clustered, unlike SVHN. Consequently it is far easier for the classifier to predict the MNIST digit as the representations do not contain any information associated with SVHN.

Table 4: Latent Space Linear Digit Classification.

| | MNIST | | | | |
|---|---|---|---|---|---|
| Model | 1.0 | 0.5 | 0.25 | 0.125 | 0.0625 |
| MEME$_\text{SVHN}$ | **0.908 ± 0.007** | 0.881 ± 0.006 | 0.870 ± 0.007 | 0.815 ± 0.005 | 0.795 ± 0.010 |
| MMVAE$_\text{SVHN}$ | 0.886 ± 0.003 | - | - | - | - |
| MVAE$_\text{SVHN}$ | 0.892 ± 0.005 | **0.895 ± 0.003** | **0.890 ± 0.003** | **0.887 ± 0.004** | **0.880 ± 0.003** |
| Ours$_\text{MNIST}$ | **0.908 ± 0.007** | 0.882 ± 0.003 | 0.844 ± 0.003 | 0.824 ± 0.006 | 0.807 ± 0.005 |
| MMVAE$_\text{MNIST}$ | 0.886 ± 0.003 | - | - | - | - |
| MVAE$_\text{MNIST}$ | 0.892 ± 0.005 | **0.895 ± 0.002** | **0.898 ± 0.004** | **0.896 ± 0.002** | **0.895 ± 0.002** |
| MEME$_\text{SPLIT}$ | **0.908 ± 0.007** | **0.914 ± 0.003** | 0.893 ± 0.005 | 0.883 ± 0.006 | 0.856 ± 0.003 |
| MVAE$_\text{SPLIT}$ | 0.892 ± 0.005 | 0.898 ± 0.005 | **0.895 ± 0.001** | **0.894 ± 0.001** | **0.898 ± 0.001** |

| | SVHN | | | | |
|---|---|---|---|---|---|
| Model | 1.0 | 0.5 | 0.25 | 0.125 | 0.0625 |
| MEME$_\text{SVHN}$ | **0.648 ± 0.012** | **0.549 ± 0.008** | **0.295 ± 0.025** | **0.149 ± 0.006** | **0.113 ± 0.003** |
| MMVAE$_\text{SVHN}$ | 0.499 ± 0.045 | - | - | - | - |
| MVAE$_\text{SVHN}$ | 0.131 ± 0.010 | 0.106 ± 0.008 | 0.107 ± 0.003 | 0.105 ± 0.005 | 0.102 ± 0.001 |
| Ours$_\text{MNIST}$ | **0.648 ± 0.012** | **0.581 ± 0.008** | **0.398 ± 0.029** | **0.384 ± 0.017** | **0.362 ± 0.018** |
| MMVAE$_\text{MNIST}$ | 0.499 ± 0.045 | - | - | - | - |
| MVAE$_\text{MNIST}$ | 0.131 ± 0.010 | 0.106 ± 0.005 | 0.106 ± 0.003 | 0.107 ± 0.005 | 0.101 ± 0.005 |
| MEME$_\text{SPLIT}$ | **0.648 ± 0.012** | **0.675 ± 0.004** | **0.507 ± 0.003** | **0.432 ± 0.011** | **0.316 ± 0.020** |
| MVAE$_\text{SPLIT}$ | 0.131 ± 0.010 | 0.107 ± 0.003 | 0.109 ± 0.003 | 0.104 ± 0.007 | 0.100 ± 0.008 |

Table 5: Correlation Values for CUB cross generations. Higher is better.

| | Image → Captions | | | | |
|---|---|---|---|---|---|
| Model | GT | $f = 1.0$ | $f = 0.5$ | $f = 0.25$ | $f = 0.125$ |
| MEME$_\text{Image}$ | 0.106 ± 0.000 | **0.064 ± 0.011** | **0.042 ± 0.005** | **0.026 ± 0.002** | **0.029 ± 0.003** |
| MMVAE$_\text{Image}$ | 0.106 ± 0.000 | 0.060 ± 0.010 | - | - | - |
| MVAE$_\text{Image}$ | 0.106 ± 0.000 | -0.002 ± 0.001 | -0.000 ± 0.004 | 0.001 ± 0.004 | -0.001 ± 0.005 |
| MEME$_\text{Captions}$ | 0.106 ± 0.000 | **0.064 ± 0.011** | **0.062 ± 0.006** | **0.048 ± 0.004** | **0.052 ± 0.002** |
| MMVAE$_\text{Captions}$ | 0.106 ± 0.000 | 0.060 ± 0.010 | - | - | - |
| MVAE$_\text{Captions}$ | 0.106 ± 0.000 | -0.002 ± 0.001 | -0.000 ± 0.004 | 0.000 ± 0.003 | 0.001 ± 0.002 |
| MEME$_\text{SPLIT}$ | 0.106 ± 0.000 | **0.064 ± 0.011** | **0.046 ± 0.005** | **0.031 ± 0.006** | **0.027 ± 0.005** |
| MVAE$_\text{SPLIT}$ | 0.106 ± 0.000 | -0.002 ± 0.001 | 0.000 ± 0.003 | 0.000 ± 0.005 | -0.001 ± 0.003 |

| | Caption → Image | | | | |
|---|---|---|---|---|---|
| Model | GT | $f = 1.0$ | $f = 0.5$ | $f = 0.25$ | $f = 0.125$ |
| MEME$_\text{Image}$ | 0.106 ± 0.000 | **0.074 ± 0.001** | **0.058 ± 0.002** | **0.051 ± 0.001** | **0.046 ± 0.004** |
| MMVAE$_\text{Image}$ | 0.106 ± 0.000 | 0.058 ± 0.001 | - | - | - |
| MVAE$_\text{Image}$ | 0.106 ± 0.000 | -0.002 ± 0.001 | -0.002 ± 0.000 | -0.002 ± 0.001 | -0.001 ± 0.001 |
| Ours$_\text{Captions}$ | 0.106 ± 0.000 | **0.074 ± 0.001** | **0.059 ± 0.003** | **0.050 ± 0.001** | **0.053 ± 0.001** |
| MMVAE$_\text{Captions}$ | 0.106 ± 0.000 | 0.058 ± 0.001 | - | - | - |
| MVAE$_\text{Captions}$ | 0.106 ± 0.000 | 0.002 ± 0.001 | -0.001 ± 0.002 | -0.003 ± 0.002 | -0.002 ± 0.001 |
| MEME$_\text{SPLIT}$ | 0.106 ± 0.000 | **0.074 ± 0.001** | **0.061 ± 0.002** | **0.047 ± 0.003** | **0.049 ± 0.003** |
| MVAE$_\text{SPLIT}$ | 0.106 ± 0.000 | -0.002 ± 0.001 | -0.002 ± 0.002 | -0.002 ± 0.001 | -0.002 ± 0.001 |

## H.2 GENERATIVE CAPABILITY

We report the mutual information between the parameters $\omega$ of a pre-trained classifier and the labels $y$ for a corresponding reconstruction $\mathbf{x}$. The mutual information gives us an indication of the amount of information we would gain about $\omega$ for a label $y$ given $\mathbf{x}$, this provides an indicator to how *out-of-distribution* $\mathbf{x}$ is. If $\mathbf{x}$ is a realistic reconstruction, then there will be a low MI, conversely, an

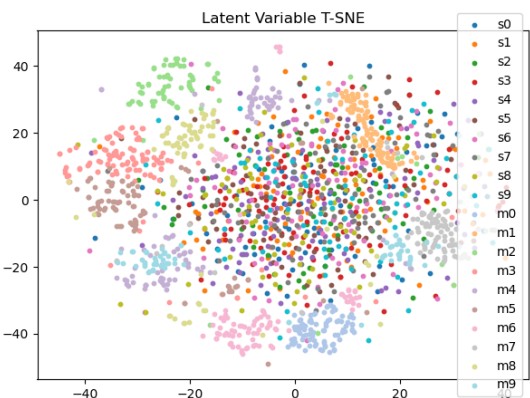

Figure 19: T-SNE plot indicating the complete failure of MVAE to construct joint representations. s indicates SVHN (low transparency), m indicates MNIST (high transparency).

un-realistic $\mathbf{x}$ will manifest as a high MI as there is a large amount of information to be gained about $\omega$. The MI for this setting is given as

$$I(y, \omega \mid \mathbf{x}, \mathcal{D}) = H[p(y \mid \mathbf{x}, \mathcal{D})] - \mathbb{E}_{p(\omega \mid \mathcal{D})} [H[p(y \mid \mathbf{x}, \omega)]].$$

Rather than using dropout Gal (2016); Smith & Gal (2018) which requires an ensemble of multiple classifiers, we instead replace the last layer with a sparse variational GP. This allows us to estimate $p(y \mid x, \mathcal{D}) = \int p(y \mid x, \omega) p(\omega \mid \mathcal{D}) \mathrm{d}\omega$ using Monte Carlo samples and similarly estimate $\mathbb{E}_{p(\omega \mid \mathcal{D})} [H[p(y \mid \mathbf{x}, \omega)]]$. We display the MI scores in Table 6, where we see that our model is able to obtain superior results.

Table 6: Mutual Information Scores. Lower is better.

| | MNIST | | | | |
|---|---|---|---|---|---|
| Model | 1.0 | 0.5 | 0.25 | 0.125 | 0.0625 |
| Ours$_{\text{SVHN}}$ | **0.075 ± 0.002** | **0.086 ± 0.003** | **0.101 ± 0.002** | **0.102 ± 0.004** | **0.103 ± 0.001** |
| MMVAE$_{\text{SVHN}}$ | 0.105 ± 0.004 | - | - | - | - |
| MVAE$_{\text{SVHN}}$ | 0.11 ± 0.00551 | 0.107 ± 0.007 | 0.106 ± 0.004 | 0.106 ± 0.012 | 0.142 ± 0.007 |
| Ours$_{\text{MNIST}}$ | **0.073 ± 0.002** | **0.087 ± 0.001** | **0.101 ± 0.001** | **0.099 ± 0.001** | **0.104 ± 0.002** |
| MMVAE$_{\text{MNIST}}$ | 0.105 ± 0.004 | - | - | - | - |
| MVAE$_{\text{MNIST}}$ | 0.11 ± 0.00551 | 0.102 ± 0.00529 | 0.1 ± 0.00321 | 0.1 ± 0.0117 | 0.0927 ± 0.00709 |
| MEME$_{\text{SPLIT}}$ | **0.908 ± 0.007** | **0.914 ± 0.003** | **0.893 ± 0.005** | **0.883 ± 0.006** | **0.856 ± 0.003** |
| MVAE$_{\text{SPLIT}}$ | 0.11 ± 0.00551 | 0.104 ± 0.006 | 0.099 ± 0.003 | 0.1 ± 0.0117 | 0.098 ± 0.005 |

| | SVHN | | | | |
|---|---|---|---|---|---|
| Model | 1.0 | 0.5 | 0.25 | 0.125 | 0.0625 |
| Ours$_{\text{SVHN}}$ | **0.036 ± 0.001** | **0.047 ± 0.002** | **0.071 ± 0.003** | **0.107 ± 0.007** | **0.134 ± 0.003** |
| MMVAE$_{\text{SVHN}}$ | 0.042 ± 0.001 | - | - | - | - |
| MVAE$_{\text{SVHN}}$ | 0.163 ± 0.003 | 0.166 ± 0.010 | 0.165 ± 0.003 | 0.164 ± 0.004 | 0.176 ± 0.004 |
| Ours$_{\text{MNIST}}$ | **0.036 ± 0.001** | **0.048 ± 0.001** | **0.085 ± 0.006** | **0.111 ± 0.004** | **0.142 ± 0.005** |
| MMVAE$_{\text{MNIST}}$ | 0.042 ± 0.001 | - | - | - | - |
| MVAE$_{\text{MNIST}}$ | 0.163 ± 0.003 | 0.175 ± 0.00551 | 0.17 ± 0.0102 | 0.174 ± 0.012 | 0.182 ± 0.00404 |
| MEME$_{\text{SPLIT}}$ | **0.648 ± 0.012** | **0.675 ± 0.004** | **0.507 ± 0.003** | **0.432 ± 0.011** | **0.316 ± 0.020** |
| MVAE$_{\text{SPLIT}}$ | 0.163 ± 0.003 | 0.165 ± 0.01 | 0.172 ± 0.015 | 0.173 ± 0.013 | 0.179 ± 0.005 |

## H.3 T-SNE PLOTS WHEN PARTIALLY OBSERVING BOTH MODALITIES

In Figure 20 we can see that partially observing MNIST leads to less structure in the latent space.

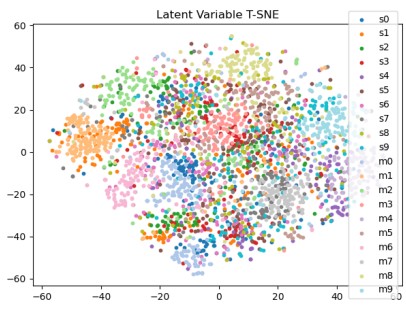 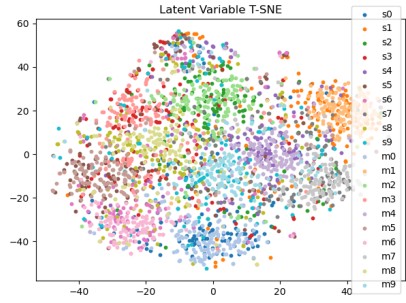

Figure 20: f = 0.25, Left) t-SNE when partially observing MNIST. Right) t-SNE when partially observing SVHN.

Table 7: Coherence Scores for MMVAE using Laplace posterior and prior.

| MNIST | SVHN |
|-------|------|
| 91.8% | 65.2% |

# I   MMVAE BASELINE WITH LAPLACE POSTERIOR AND PRIOR

The difference in results between our implementation of MVAE and the ones in the paper (Shi et al., 2019), is becuase we restrict MEME to use Gaussian distributions for the posterior and prior, and therefore we adopt Gaussian posteriors and priors for all three models to ensure like-for-like comparison. Better results for MMVAE can be obtained by using Laplace posteriors and priors, and In Table 7 we display coherence scores using our implementation of MMVAE using a Laplace posterior and prior. Our implementation is inline with the results reported in Shi et al. (2019), indicating that the baseline for MMVAE is accurate.

# J   ABLATION STUDIES

Here we carry out two ablation studies to test the hypotheses: 1) How sensitive is the model to the number of pseudo samples in $\lambda$ and 2) What is the effect of training the model using only paired data for a given fraction of the dataset.

## J.1   SENSITIVITY TO NUMBER OF PSEUDO-SAMPLES

In Figure 21 we plot results where the number of pseudo samples is varied for different observation rates. Ideally we expect to see the results decrease in their performance as the number of pseudo-samples is minimised. This is due to the number of components being present in the mixture $p_{\lambda^{\mathbf{t}}}(\mathbf{z}) = \frac{1}{N} \sum_{i=1}^{N} p_\psi(\mathbf{z} \mid \mathbf{u}_i^{\mathbf{t}})$, also being decreased, thus reducing the its ability to approximate the true prior $p(\mathbf{z}) = \int_{\mathbf{t}} p_\psi(\mathbf{z} \mid \mathbf{t}) p(\mathbf{t}) dt$. As expected lower observation rates are more sensitive, due to a higher dependence on the prior approximation, and a higher number of pseudo samples typically leads to better results.

## J.2   TRAINING USING ONLY PAIRED DATA

Here we test the models ability to leverage partially observed data to improve the results. If the model is successfully able to leverage the partially observed samples, then we should see a decrease in the efficacy if we train the model using only paired samples, i.e. a model trained with 25% paired and 75% partially observed should perform improve the results over a model trained with only the 25% paired data. In other words we omit, the first two partially observed terms in (5), discarding $\mathcal{D}_{\mathbf{s}}$ and $\mathcal{D}_{\mathbf{t}}$. In Figure 22 we can see that the model is able to use the partially observed modalities to improve its results.

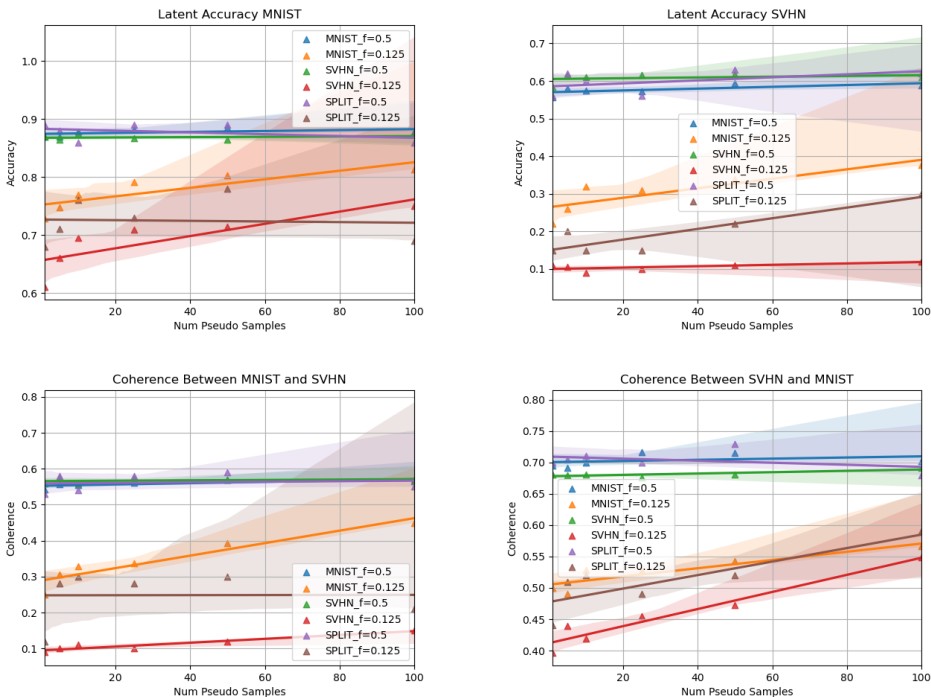

Figure 21: How performance varies for different numbers of psuedo samples. Number of pseudo samples ranges from 1 to 100 on the x axis.

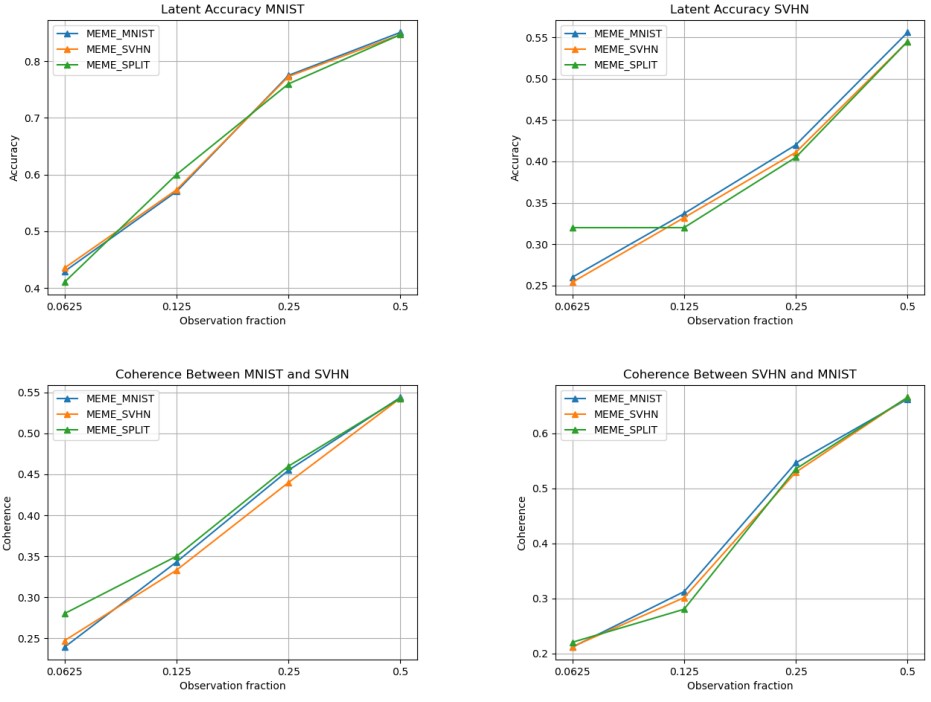

Figure 22: How performance varies when training using only a fraction of the partially observed data.

# K  TRAINING DETAILS

The architechtures are very simple and cean easily be implemented in popular deep learning frameworks such as Pytorch and Tensorflow. However, we do provide a release of the codebase at the following location: https://github.com/thwjoy/meme.

**MNIST-SVHN**  We provide the architectures used in Table 8b and Table 8a. We used the Adam optimizer with a learning rate of $0.0005$ and beta values of $(0.9, 0.999)$ for 100 epochs, training consumed around 2Gb of memory.

**CUB**  We provide the architectures used in Table 8c and Table 8d. We used the Adam optimizer with a learning rate of $0.0001$ and beta values of $(0.9, 0.999)$ for 300 epochs, training consumed around 3Gb of memory.

| Encoder | Decoder |
|---|---|
| Input $\in \mathbb{R}^{1x28x28}$ | Input $\in \mathbb{R}^L$ |
| FC. 400 ReLU | FC. 400 ReLU |
| FC. $L$, FC. $L$ | FC. 1 x 28 x 28 Sigmoid |

(a) MNIST dataset.

| Encoder |
|---|
| Input $\in \mathbb{R}^{1x28x28}$ |
| 4x4 conv. 32 stride 2 pad 1 & ReLU |
| 4x4 conv. 64 stride 2 pad 1 & ReLU |
| 4x4 conv. 128 stride 2 pad 1 & ReLU |
| 4x4 conv. L stride 1 pad 0, 4x4 conv. L stride 1 pad 0 |

| Decoder |
|---|
| Input $\in \mathbb{R}^L$ |
| 4x4 upconv. 128 stride 1 pad 0 & ReLU |
| 4x4 upconv. 64 stride 2 pad 1 & ReLU |
| 4x4 upconv. 32 stride 2 pad 1 & ReLU |
| 4x4 upconv. 3 stride 2 pad 1 & Sigmoid |

(b) SVHN dataset.

| Encoder | Decoder |
|---|---|
| Input $\in \mathbb{R}^{2048}$ | Input $\in \mathbb{R}^L$ |
| FC. 1024 ELU | FC. 256 ELU |
| FC. 512 ELU | FC. 512 ELU |
| FC. 256 ELU | FC. 1024 ELU |
| FC. $L$, FC. $L$ | FC. 2048 |

(c) CUB image dataset.

| Encoder |
|---|
| Input $\in \mathbb{R}^{1590}$ |
| Word Emb. 256 |
| 4x4 conv. 32 stride 2 pad 1 & BatchNorm2d & ReLU |
| 4x4 conv. 64 stride 2 pad 1 & BatchNorm2d & ReLU |
| 4x4 conv. 128 stride 2 pad 1 & BatchNorm2d & ReLU |
| 1x4 conv. 256 stride 1x2 pad 0x1 & BatchNorm2d & ReLU |
| 1x4 conv. 512 stride 1x2 pad 0x1 & BatchNorm2d & ReLU |
| 4x4 conv. L stride 1 pad 0, 4x4 conv. L stride 1 pad 0 |

| Decoder |
|---|
| Input $\in \mathbb{R}^L$ |
| 4x4 upconv. 512 stride 1 pad 0 & ReLU |
| 1x4 upconv. 256 stride 1x2 pad 0x1 & BatchNorm2d & ReLU |
| 1x4 upconv. 128 stride 1x2 pad 0x1 & BatchNorm2d & ReLU |
| 4x4 upconv. 64 stride 2 pad 1 & BatchNorm2d & ReLU |
| 4x4 upconv. 32 stride 2 pad 1 & BatchNorm2d & ReLU |
| 4x4 upconv. 1 stride 2 pad 1 & ReLU |
| Word Emb.$^T$ 1590 |

(d) CUB-Language dataset.

Table 8: Encoder and decoder architectures.

