# OpenReview forum: "Learning Multimodal VAEs through Mutual Supervision"
_ICLR.cc/2022/Conference — ICLR 2022 Spotlight_

### Official Review · Reviewer_WKus · 2021-11-02

**Correctness:** 4
**Technical Novelty And Significance:** 3
**Empirical Novelty And Significance:** 3
**Recommendation:** 6
**Confidence:** 3

**Main Review:**

Strengths:
1. The presented framework (MEME) is an interesting and technically sound method for multi-modal data modeling. The paper is well-structured and easy to follow.
2. The authors conduct extensive experiments and employ many different metrics to demonstrate the effectiveness of the methods. The experimental results are good enough to make a contribution to multi-modal VAEs.
3. The method can handle partial data modalities. Even though the idea behind is not novel (like the partial label of semi-supervised VAEs), I still consider it as a contribution, since prior work usually handles the partially-observed modalities in a hard way (zero-padding or discarding).

Weakness:
1. The method extends the CCVAE which models the dependency between label and data. The information in labels is definitely not comparable to the information in a data modality. I agree that mutual supervision can mitigate this issue as it establishes a symmetric information flow. However, in qualitative results (Fig.12 - Fig.17) it is quite obvious that the generated samples of SVHN (left) are limited in certain colors (grayish) and shapes (thin and centered), while MNIST generations (right) contain much more variations. Can the authors provide more details on the architecture and parameters related to each modality? Can more powerful decoders for SVHN help generate more realistic SVHN samples?

2. I'm a bit concerned about the scalability of the method. Even though the authors provide some potential ways to scale the current bi-modal setting to a multi-modal setting. Unlike other multi-modal VAEs that simply concatenates a new modality, MEME is based on a bi-modal setting. Thus three modalities will already need 6 information flows (e.g. for ($a,b,c$) modalities, $a \rightarrow b, b \rightarrow a, a \rightarrow c, c \rightarrow a, b \rightarrow c, c\rightarrow b$). Could the authors elaborate more on how to scale MEME to modalities > 2? The authors can also add some results on 3-modal datasets like MNIST-SVHN-Text used in [1] to compare the performance and training time of MEME and other baselines.


Minor comments:
1. The authors need to double-check the manuscripts --- some typos I can find may confuse the audience: 1) In  Fig.6 MEME_Caption should be MEME_Image? 2) in Fig.9 caption the middle is MMVAE, not MVAE?

2. I'm curious about the comb-like distribution of Wassertein distance in Fig.8 SVHN-MNIST. Do authors have any intuition or comments on it?

[1] Thomas M. Sutter, Imant Daunhawer, Julia E. Vogt, Multimodal Generative Learning Utilizing, Jensen-Shannon-Divergence, NeurIPS 2021

**Summary Of The Paper:**

The authors propose a novel approach based on Variational Autoencoders to model the joint distribution of heterogeneous data (perceptually multi-modal, e.g. vision + language). Unlike prior work that typically handles idiosyncratic modalities with the explicit combination (concatenation or factorization), this proposed method implicitly introduces a dependency between two modalities via a prior regularization. The method extends CCVAE (Joy et al. 2021) by replacing the label as another modality. To compensate for the information disparity from the label to a data modality, the authors present a "mutual supervision" that uses a bi-directional information flow. The method can also train with partially-observed data where some modalities are missing.

**Summary Of The Review:**

This paper presents a method extending CCVAE from label and data to multiple modalities. The way of CCVAE handling semi-supervised can be used for training with partially-observed data. Even though the novelty is somewhat limited in VAE literature, it's still a contribution to multi-modal generative modeling. Thus I vote for a borderline acceptance at this point.

---

> ### Author Response · Authors · 2021-11-12
> **Response to Reviewer**
>
> Thank you for your response and noting the merits of the paper, we particularly appreciated your acknowledgments of the experiments.
>
> Architectures are present in the Appendix. We would put these results down to typical generations from a VAE, which tend to be noisier and have a lower fidelity and are evident in MVAE and MMVAE. It is possible that much more powerful decoders could be used such as autoregressive  or hierarchical (Kingma et.al. 2016), however wielding such complex networks possess significant challenges in its own right, so here we explicitly just chose to pay attention to the methodological aspect. However as pointed out in http://ruishu.io/2017/01/14/one-bit/ more powerful decoders can take information away from the latent space.
>
> We have provided a general comment to extending beyond the bi-modal case above.
>
> ### Minor comments
>
> 1. These are correct, we will update accordingly.
>
> 2. The comb like distribution is likely caused by the individual classes. We will investigate further and confirm this for you.

---

> ### Comment · Reviewer_WKus · 2021-11-28
> **Response to authors**
>
> Thanks for the response and effort to address my concern. Though I value the contribution of the empirical performance, I agree with other reviewers that the novel is limited. I will keep my rating as borderline accept.

---

### Official Review · Reviewer_1avA · 2021-11-02

**Correctness:** 4
**Technical Novelty And Significance:** 3
**Empirical Novelty And Significance:** 2
**Recommendation:** 8
**Confidence:** 4

**Main Review:**

Strengths:
1) The main strength of this paper lies in its novel formulation of multimodal VAES and it avoids the need for explict specifications such as products and regularizers to combine information across modalities. The proposed approach also allows the joint distribution to be modeled under the partially-observed setting, where a percentage of observations have missing modalities.  The approach is also intuitive and well-detailed such that it is easily understandable.

2) The equations are easy to understand and constructing a mirrored version of the VAE by swapping the modalities as well as their corresponding parameters is a clever idea that is premised on  the insight of finite conditional entropies.

3) The authors empirically prove the effectiveness of their proposed approach through extensive experiments and ablations on the
MNIST-SVHN and CUB datasets.

Weaknesses:

1) The reasons for confining the focus of this work to two modalities is completely understandable. However, given the fact that it works well under the partially observed setting, it will be insightful to expand the discussion on generalizing beyond two modalities. With the recent focus on representation learning from multiple (more than 2) modalities, it might be helpful to discuss more if it's possible to extend the implicit combination to multiple modalities instead of using explicit combinations.

2) It may also be helpful to provide more explanation on the intuition underlying the use of learnable pseudo samples to estimate the prior for learning from partially observed data.

3) Minor point: the current results in the main paper are reflected through figures. It would be nice to have absolute numbers to compare in the main paper. For example, table 3 could be included in the main paper to highlight the effectiveness of the approach, despite the space constraints.

**Summary Of The Paper:**

This paper presents an approach to model the joint distribution over data from different modalities such as vision and language. It seeks to address a limitation that is common to prior work on multimodal VAEs, which have typically combined information across modalities by using explicit methods including products, mixtures and their combinations.  Specifically, this work proposes to exploit mutual supervision between different modalities to circumvent the need for the above-mentioned explicit combinations. Additionally, it provides a natural and intuitive extension to learning from partially observed data. The authors empirically prove the effectiveness of their proposed approach over prior work under both partial and complete data observation settings.

**Summary Of The Review:**

In summary, this paper presents an interesting and intuitive approach for modeling the joint distributions across modalities. In particular, it appears to work well in the partially observed setting. Given the nature of real-world data which is usually very noisy, this is significant. Coupled with informative figures and extensive experiments (including those in the supplementary material), this paper could serve as a very useful reference for modeling the joint distribution of all modalities concurrently. Hence, I recommend acceptance.

---

> ### Author Response · Authors · 2021-11-12
> **Response to Reviewer**
>
> Thank you for your response and noting the merits of the paper, particularly with the explanations of the bi-modal case.
>
> We have added a general comment on the extension beyond the bi-modal above.
>
>
> Essentially we are now treating the prior as a mixture of conditional priors. The motivation for this comes from the fact we wish to marginalize the unobserved random variable $p(z) = \int p(z, t)dt$, which is intractable. Consequently, we wish to approximate this integral as mixture over the dataset $p(z) \approx \sum_n p(z\mid t_n) p(t_n)$, which we further approximate using the learned pseudo samples similar to Tomczak & Welling 2017.
>
>
> The tabularised results are in the Appendix, we have added a pointer in the updated manuscript. If the reviewer feels it will strengthen the paper, we can look into adding the tables into the main paper.

---

### Official Review · Reviewer_ELno · 2021-11-03

**Correctness:** 4
**Technical Novelty And Significance:** 2
**Empirical Novelty And Significance:** 2
**Recommendation:** 5
**Confidence:** 3

**Main Review:**

Strengths

1. The paper is well written and this reviewer enjoyed reading it.

2. The proposed approach is solid and interesting
2. The authors show rich experiments and analyses, including ablation studies and relatedness. Also, they ran the same models multiple times with different seeds and showed the deviation on the plots.


Weaknesses

1. In “Generalisation beyond two modalities” (on page 5), the authors say it is quite straightforward to extend MEME to be used for more than two modalities. Still, this reviewer can imagine it would be quite a complicated system with more than two modalities. Can you explain more details about the extension?

2. Unclearness in Section 4.

2-1. In Section 4, the notations with subscripts are confusing. Does the subscript SPLIT indicate when equal amounts of unpaired data from two modalities (e.g. MNIST and SVHN) are used?

2-2. Although the authors said in Sec. 4.2 that the MEME contains rich class information from inputs, in Figure 7, the left plot does not support that the MEME models have a stronger representative power than the MVAE models, when f < 1, except the MEME_SPLIT model. This reviewer is unsure why the MEME models that show a high performance in the cross coherence metric shows lower performance in the classification task.

2-3. In Figure 7 (left), MVAE_SPLIT performs better when less observation is available than seeing the entire data. This phenomenon also happens in MEME_SPLIT in the Latent Accuracy SVHN plot. Why?

2-4. In Figure 8 (bottom), the orange portion seems larger than the blue portion in the MEME plot, while in the other two plots at the bottom, the numbers of unpaired and paired samples seem equal. This review is curious if the different settings were used for the MEME models.

2-5. In Figure 9 (top, right), the MMVAE model completely fails to extract class information. However, this result is not coincident with the results from Figure 5, which shows a lot higher scores than MMVAEs’


3. Typos.

3-1. On page 3 (right before Sec. 3.2) Θ = {θ, θ} ← there are two thetas.

3-2. In Figure 6, the subscripts are {Caption, Sentence, SPLIT}. Doesn't either Caption or Sentence have to be replaced with Image?


4. The paper introduced a novel approach to the mirrored semi-supervised VAE by adding two additional regularizations to solve the missing unpaired data problem in the multimodal setting. However, the technical novelty is limited, except for extending the work of a semi-supervised VAE [Joy et al. (2021)].  Also, the problem itself is not new as the multimodal generative models were introduced in prior work [Wu et al. (2018)] and [Shi et al. (2019)].

**Summary Of The Paper:**

The paper solves the multimodal problem in the partially-observed setting. With the observation that no one modality has all the information, the authors proposed a novel approach called MEME, which symmetrizes the semi-supervised VAE formulation by constructing a mirrored version. The proposed approach was demonstrated on standard metrics (cross coherence, latent accuracy) for multimodal VAEs across both partial and complete data settings (MNIST-SVHN, CUB) and shows that it outperforms prior work on both.

**Summary Of The Review:**

The paper is about the new approach to solve the missing unpaired data problem in the multimodal setting. Although the approach is solid and interesting, it is incremental to the prior work. Some of the experiments demonstrating the superiority of the proposed approach are also unclear.

---

> ### Author Response · Authors · 2021-11-12
> **Response to Reviewer**
>
> Thank you for your response and noting the merits of the paper, we particularly appreciated your acknowledgments of the experiments.
>
> 1. We have added a general comment above on the extension beyond the bi-modal case.
>
> 2.1: Correct, we will clarify further
>
> 2.2: Indeed, we also thought this was rather interesting. We hypothesise that this is due to MVAE learning to place embeddings for the same class in separate regions of the latent depending on the modality. E.g. when there is low supervision rates, the model can essentially learn to place SVHN in one regions and MNIST in the other. This factorisation enables reconstruction and allows simple classifiers to be learnt on the latent space, but because the modalities are placed in different regions it suffers severely when performing cross generation.
>
> 2.3: It is worth highlighting that there will be a certain amount of uncertainty when performing this experiment as we are learning a classifier on top of a stochastic data-set. Consequently, it is entirely possible that this effect either comes from a better classifier and a latent space which is slightly more suitable to be classified.
>
> 2.4: We believe this might be deceptive, equal numbers of paired and unpaired samples were used when generating these plots. We used exactly the same settings across models.
>
> 2.5: Apologies, this is typo. The plot in the middle represents MMVAE and the one on the right MVAE. It should read: Distance matrices for KL divergence between classes for SVHN and MNIST (Top) and dendrogram (Bottom) for: Ours (Left), MMVAE (middle) and MVAE (Right).
>
> 3. Thank you for pointing out these typos, they will be amended in the updated manuscript.

---

### Official Review · Reviewer_ag7M · 2021-11-03

**Correctness:** 3
**Technical Novelty And Significance:** 3
**Empirical Novelty And Significance:** 3
**Recommendation:** 6
**Confidence:** 5

**Main Review:**

Strengths:
- The paper is very well organized and easy to read.
- The proposed MEME is a straightforward extension of an existing semi-supervised VAE to a multimodal setting, which is easy to understand and can be easily adapted to partial train settings.
- Experiments confirm the effectiveness of MEME compared to MVAE and MMVAE in both partial and complete data settings.

Weakness:
- Since MEME is an extension of existing semi-supervised models, it is understandable that MEME performs better than MVAE in partial settings. However, why does MEME, which is just an extension of semi-supervised VAEs, perform better than MVAE and MMVAE even in the complete setting? What parts of MEME are important to achieve better results than them? In other words, the authors discuss the merits of MEME only from a semi-supervised perspective, and therefore should explicitly discuss why it is good in the complete setting. Also, the authors do not explain why MEME shows a better trend than MVAE or MMVAE in the relatedness experiments.
- I understand that the authors are specifically focusing on the bimodal case in this paper. However, traditional explicit combinations, such as mixture and products, are intended to combine more than two modalities. In the last part of section 3.3, the authors state that an explicit combination can be added on top of the implicit combination to handle cases beyond two modalities (I don't understand the details of this method. Does this mean that the bimodal representations by the implicit combination are combined with the explicit one?).  However, this approach is a combination of implicit and explicit methods, so it does not mean that implicit itself can be extended to more than two modalities. Therefore, this implies that implicit combinations are essentially only applicable to bimodal. It is true that early multimodal VAEs such as JMVAE only considered bimodal, but unlike MEME, it does not mean that it cannot be extended to more than two modalities, although it requires exponentially increasing memory cost. Therefore, the fact that MEME is essentially not scalable to more than two is a fatal problem in the study of multimodal VAEs, which is nevertheless underestimated and hardly discussed in this paper.
- Even though MoPoE is an extension of MVAE and MMVAE and has been shown to perform better than them, why hasn't it been compared in experiments?

**Summary Of The Paper:**

This paper proposes MEME as a new method of multimodal VAE, which is an extension of semi-supervised VAEs and can handle partial train settings. Experimental results show that the proposed method outperforms the conventional methods, MVAE and MMVAE, in both partial and complete settings. Furthermore, the proposed method shows an interesting trend in its ability to capture the relatedness compared to these.

**Summary Of The Review:**

The MEME proposed by the authors in this study achieves high performance in both partial and complete cases. It also shows interesting results in terms of relatedness. However, there is a lack of discussion on why MEME is better than existing methods and an explanation of its limitations.

---

> ### Author Response · Authors · 2021-11-12
> **Response to Reviewer**
>
> We would like to submit that MEME should not be understood as simply an application of semi-supervised VAEs to multi-modal representation learning. Indeed, the problem here is fundamentally different in that different modalities usually contain idiosyncratic information. This is in contrast to semi-supervised learning where _one_ modality is paired with a label that does not contain any information that cannot in principle be retrieved from that same modality.
> With MEME the aim is here to discuss the _techniques_ (such as those relying on information flow) that have been successfully applied to semi-supervised settings so as to apply them to multi-modal VAEs. It is immediate, however, a straightforward application of semi-supervised models to data modalities with unique characteristics is not possible and any successful approach will come with methodological and engineering challenges that are not present in semi-supervised approaches.
> Thus, while the handling of missing modalities is a clear benefit of this approach, our method should not be reduced to an application of semi-supervised VAEs.
>
> In short, the core of the idea lies in the bi-directional _information flow_ which we term “mutual supervision” which is of relevance in supervised _and_ semi-supervised settings. We, therefore, do not see a reason to expect better performance only in the semi-supervised setting. Why our approach might outperform particular alternatives is discussed below:
>
> - Outperforming MVAE is perhaps not that surprising given that, in practice, products are quite hard to train effectively, as also observed by Shi et. al., 2019. Moreover, for the MVAE, the 'complete' setting actually requires artificial subsampling in order to ensure the product distribution can actually be used effectively at test time.
> - Outperforming MMVAE is indeed interesting, and we believe this is due to the use of a less-constrained regulariser (i.e., the KL between posterior and prior) than a (uniformly weighted) mixture.
> Note that in both MEME and MMVAE, the decoders are trained using samples from _both_ encoders---through the symmetric sum in MEME, and through the mixture in MMVAE.
> - Why MEME shows an improvement in terms of relatedness is directly due to the mutual supervision in MEME’s training regime. Here MEME directly regularises the encoding distributions from each modality to be similar for the same class. Unlike for MVAE and MMVAE, where the only pressure for them to be similar comes from the reconstruction loss.
>
> We’ve added a general comment on how to extend the model beyond the bi-modal case.
>
> We omitted MoPoE due to the lack of focus on the partially observed case.

---

### Author Response · Authors · 2021-11-12
**Extension Beyond the Bi-Modal Case**

Extension beyond the bi-modal case.

We would like to offer some context to how MEME could be extended for the case when modalities M > 2.

Note that the central thesis in MEME is that the evidence lower bound (ELBO) offers an implicit way to regularise different representations if viewed from the posterior-prior perspective, which can be used to build effective multimodal DGMs that are additionally applicable to partially-observed data.

In MEME, we explore the utility of this implicit regularisation in the simplest possible manner to show that a direct application of this to the multi-modal setting would involve the case where M = 2.

The way to extend, say for M = 3, involves additionally employing an explicit combination for two modalities in the prior (instead of just 1). This additional combination could be something like a mixture or product, following from previous approaches.

More formally, if we were to denote the implicit regularisation between posterior and prior as R_i(., .), and an explicit regularisation function R_e(., .), and the three modalities as m_1, m_2, and m_3, this would mean we would compute

1/3 * [ R_i(m_1, R_e(m_2, m_3)) + R_i(m_2, R_e(m_1, m_3)) + R_i(m_3, R_e(m_1, m_2)) ]

assuming that R_e was commutative.

There are indeed more terms to compute now compared to M=2, which only needs R_i(m_1, m_2), but note that R_i is still crucial---it does not diminish because we are additionally employing R_e.

As stated in prior work (e.g. MVAE/MMVAE/JMVAE), we follow the reasoning that the actual number of modalities, at least when considering embodied perception, is not likely to get much larger, so the increase in number of terms, while requiring more computation, is unlikely to become intractable. Note that prior work on multimodal VAEs (e.g. JMVAE, MVAE, MMVAE, etc) also suffer when extending the number of modalities in terms of the number of paths information flows through.

We do not explore this setting empirically as our priary goal is to highlight the utility of this implicit regularisation for multi-modal DGMs, and its effectiveness at handling partially-observed data.

---

### Author Response · Authors · 2021-11-23
**Thank you and updated manuscript**

We would like to thank the reviews for their hard work in providing high-quality reviews in such a short time frame and noting the merits of the paper, such as the quality of the experiments. We have uploaded the latest draft addressing your concerns and adding additional information where required. e.g. extension beyond the bi-modal case and t-sne plots for MVAE.

---

### Decision · Program_Chairs · 2022-01-20

**Decision:**

Accept (Spotlight)

**Comment:**

PAPER: This paper introduces a new method to learn joint representations from multimodal data, with potentially missing data. The primary novelty builds from the idea of semi-supervised VAE, introducing the concept of bi-directional information flow, which is termed “mutual supervision”. This approach brings the same advantages of semi-supervised VAE to the multimodal setting, allowing the cross-modal interactions to be modeled in the latent space.
DISCUSSION: The discussion brought many important issues, addressed by both reviewers and authors. In general, it seems that most reviewers appreciate the technical novelty of the paper, related to the mutual supervision. While some concerns were expressed about the similarity with semi-supervised VAE (Joy et al., 2021), I would agree with other reviewers and the authors that the extension is not straightforward. Bi-directional information flow is a worthwhile novelty in itself. One reviewer also mentioned a concern about previous work on multimodal generative models; previous work on the same topic should not preclude new papers, as long new technical ideas are proposed. The final observation is about modeling more than 3 modalities. This is effectively a challenge with the proposed idea and should be acknowledged in the paper, but it is also an issue for many other approaches. New research will be needed to study 3+ modalities, but it should be seen as a future work direction.
SUMMARY: Based on the reviews, discussion and personal reading of the paper, I lean towards acceptance of this paper. The paper introduces a new technical idea (bi-directional information flow, aka mutual supervision) which enables multimodal representation learning with missing data. The authors should revise their paper to acknowledge potential limitations of the approach (e.g., complexity challenges with 3+ modalities), but the idea is very interesting and worth publication.